# Protective effect and mechanism of *Lacticaseibacillus paracasei* 207-27 administration on colitis in antibiotic-exposed mice in early life

Niya Li,[1] Fengling Jiang,[2] Yunyi Wang,[1] Xiaolin Xu,[1] Liang Li,[3] Xiaolei Ze,[3] Huijing Liang,[1,3] Ruyue Cheng,[1] Fang He,[1] Xi Shen[1]

**ABSTRACT** Evidence shows that antibiotic exposure in early life increases the risk of inflammatory bowel disease by causing gut microbiota dysbiosis and affecting intestinal immune system development. This study aimed to investigate the possible protective effect and mechanism of short-term and long-term applications of *Lacticaseibacillus paracasei* 207-27 for ameliorating gut microbiota disorders due to early-life antibiotic exposure in mice with dextran sulfate sodium (DSS)-induced colitis. We intervened neonatal mice with ceftriaxone, and *L. paracasei* 207-27 was administered 2 h following antibiotic administration. *L. paracasei* 207-27 was used for continuous gavage for 3 and 6 weeks. The first batch of mice was sacrificed on day 21, and the second batch was sacrificed on day 46 following a 4-day intervention with 3% DSS to induce colitis. Results showed that on day 21, the ceftri + 207-27 group had higher alpha diversity of the gut microbiota, short-chain fatty acid levels, and splenic immune factor levels than the ceftri group, with significantly lower serum interleukin-6 (IL-6) and tumor necrosis factor-α (TNF-α) levels. On day 46, the 207-27 short-term group had similarly lower splenic and serum IL-6 and TNF-α levels than the DSS group. Regarding colonic IL-17, TNF-α, and interferon-γ (INF-γ) levels, a decreasing trend was observed in the 207-27 short-term and long-term groups compared with that in the ceftri–DSS group, and the 207-27 long-term group had significantly higher claudin and secretory immunoglobulin A levels. In conclusion, short-term *L. paracasei* 207-27 administration improves gut microbiota composition and effectively alleviates colitis symptoms, and long-term *L. paracasei* 207-27 administration protects the intestinal barrier.

**IMPORTANCE** This study is the first to apply *Lacticaseibacillus paracasei* 207-27 on colitis in antibiotic-exposed mice in early life and observe the protective effect. Short-term *L. paracasei* 207-27 administration improves the symptoms of colitis, whereas long-term *L. paracasei* 207-27 administration promotes intestinal barrier function.

**KEYWORDS** *Lacticaseibacillus paracasei* 207-27, colitis, antibiotic, early life

Inflammatory bowel disease (IBD), characterized by chronic recurrent inflammation involving the whole colon, can be classified as either Crohn's disease or ulcerative colitis (1, 2). IBD symptoms include body weight loss, bloody diarrhea, ulcers of the epithelial cells, inflammatory cell infiltration, crypt edema, colon shortening, and microbiota alteration (2). The incidence rate of IBD has been increasing globally, which causes a huge worldwide healthcare burden (3). Current evidence indicates that several key factors, including altered microbial composition, genetic factors, defects in mucosal barrier function, and innate and adaptive immunity, contribute to IBD initiation and progression (4). Nowadays, microbial-targeted therapies for restoring gut microbial homeostasis offer an alternative to IBD treatment (5).

**Peer Reviewer** Michael G. Ganzle, University of Alberta, Edmonton, Alberta, Canada

Address correspondence to Xi Shen, hxgwshenxi@sina.com.

Niya Li and Fengling Jiang contributed equally to this article. The author order was determined by drawing straws.

The authors declare no conflict of interest.

See the funding table on p. 16.

Probiotics are live microorganisms that, when administered in adequate amounts, confer health benefits on the host (6). Substantial evidence exists that the administration of *Lacticaseibacillus paracasei* strains can alleviate the symptoms of dextran sulfate sodium (DSS)-induced colitis in mice by improving immunomodulation and changing the gut microbiota composition, including *L. paracasei* R3 (*L.p* R3) (7), *L. paracasei* MSMC39-1 (8), and *L. paracasei subsp. paracasei* NTU 101 (NTU 101) (9). In our previous study, *L. paracasei* 207-27 was isolated from healthy infant feces, and we noted that *L. paracasei* 207-27 had potential anti-inflammatory and anti-allergic effects (10–12). We hypothesize that *L. paracasei* 207-27 can also alleviate symptoms by playing an immunomodulatory role in the DSS-induced murine colitis model.

Antibiotics are the most frequently prescribed drugs in neonates, with 8% of all infants in Europe exposed to antibiotics in the first week of life (13). Early-life antibiotic treatments influence a large fraction of the global population and are associated with global epidemic health problems, including immunological diseases (14). Early life is a significant period of susceptibility for IBD development later in life (15). Several retrospective clinical studies have shown that antibiotic exposure in early life increases IBD risk by causing gut microbiota dysbiosis and affecting host immune system development (16). The early-life microbiome plays a central role in priming the immune system (17). Therefore, we hypothesized that probiotics application can mitigate the effects of antibiotic exposure in early life.

Well-defined probiotics may become an alternative therapy for IBD. We here analyzed the role of short-term and long-term *L. paracasei* 207-27 administration in DSS-induced colitis under conditions of antibiotic exposure in early life. Therefore, we investigated how the probiotic *L. paracasei* 207-27 improves antibiotic-induced intestinal destruction by regulating the immune system and gut microbiota composition and preventing murine colitis.

## MATERIALS AND METHODS

### *Lacticaseibacillus paracasei* 207-27 preparation

*Lacticaseibacillus paracasei* 207-27 (*L. paracasei* 207-27, which is also named LPB27 in its commercialized product) was originally isolated from fresh feces of healthy infants born in China (12) and was deposited at Guangdong Microbial Culture Collection Center (GDMCC) under the Budapest Treaty, with deposit code GDMCC 60960. To confirm the bacterial strain at the species level, V3–V4 of 16S rRNA sequencing was amplified and sequenced. Bacteria were cultured on de Man, Rogosa, and Sharpe agar (Land Bridge, Beijing, China) under an anaerobic atmosphere at 37°C for 48 h and passaged three times before use. Cultures were collected, washed twice with sterile saline, and subsequently diluted to a suitable concentration according to the colony-forming units (CFU)-absorbance standard curve.

### Mice and treatment

Fourteen-day-old timed-pregnant BALB/C female mice, purchased from Liaoning Changsheng Biotechnology Co., Ltd. (Liaoning, China), were housed under standard specific pathogen-free laboratory conditions (approval number: SYXK2023-0011). As shown in Fig. S1, on days 1–21, neonatal mice ($n = 42$) were randomly selected and divided into the following three groups: NS group ($n = 18$), ceftri group ($n = 12$), and ceftri + 207-27 group ($n = 18$). The mice were housed with a feeding mother at a density of six pups per cage, resulting in three cages for the NS group, two cages for the ceftri group, and three cages for the ceftri + 207-27 group. Saline, ceftriaxone sodium (100 mg/kg), and ceftriaxone sodium (100 mg/kg) + *L. paracasei* 207-27 ($1 \times 10^7$ CFU/day on 1–7 days, $1 \times 10^8$ CFU/day on 8–14 days, and $1 \times 10^9$ CFU/day on 15–21 days; probiotics should be administered 2 h following antibiotic administration) were respectively administered for a 3-week continuous gavage. After weaning at the end

of the intervention on day 21, six mice in each group were randomly selected as the first batch of mice to be sacrificed. Two mice per cage in the NS group were randomly sacrificed, and the remaining mice continued to be divided into NS-water ($n = 6$) and NS–DSS groups ($n = 6$), at which time there were six mice in each group (two mice/cage, three cages); three mice per cage in the NS-ceftri group were randomly sacrificed, and the remaining mice were in the ceftri–DSS group ($n = 6$) (three mice/cage, two cages); and three mice per cage in the ceftri + 207-27 group were randomly sacrificed with two mice per cage, and the remaining mice continued to be divided into 207-27 short-term ($n = 6$) and 207-27 long-term groups ($n = 6$) (two mice/cage, three cages). The 207-27 long-term group was continuously gavaged at $1 \times 10^9$ CFU/day for 3 weeks, and the rest were fed without any intervention until day 42. On day 43, except for the NS-water group, all mice in the other groups were administered 3% DSS to induce colitis. On day 46, the experiment ended, and the remaining mice were sacrificed as the second batch.

At the end of the experiment, fresh feces were collected and frozen at −80℃. The body weight of each group of mice was measured daily until day 21 and weekly after day 21 to calculate the growth multiplier. On day 21, fresh feces were collected from all mice and weekly thereafter. Blood samples were collected from the first batch of mice on day 21 and from the second batch of mice on days 42 and 46. The small intestine, colonic tissue, and cecum contents were collected from both batches of mice.

## Colon inflammation assessment

Colonic tissues were collected from the sacrificed mice, fixed in 10% neutral buffered formalin for 48 h, and subsequently dehydrated with ethanol and embedded in paraffin. Sections were stained with hematoxylin and eosin (H&E) for histopathological analysis. A professional pathology teacher evaluated the sample sections.

Inflammatory pathology in colonic tissues was scored according to the following criteria: inflammation (0 = none, 1 = slight, 2 = moderate, and 3 = severe); crypt damage (0 = none, 1 = basal 1/3, 2 = basal 2/3, 3 = crypt loss with surface epithelium intact, and 4 = crypt and surface epithelial destruction); the percentage of the area involved according to inflammation (1 = 1%–25%, 2 = 26%–50%, 3 = 51%–75%, and 4 = 76%–100%); the percentage of crypt damage (1 = 1%–25%, 2 = 26%–50%, 3 = 51%–75%, and 4 = 76%–100%); and the depth of inflammation (0 = normal, 1 = mucosa, 2 = submucosa, and 3 = transmural). The final pathology score of the colonic tissue was the sum of the component scores (18, 19).

## Fecal microbiome analysis

Fresh feces from mice were collected on days 21 and 46 and subsequently frozen at −80℃. Following the manufacturer's instructions, bacterial genomic DNA was extracted from the colonic contents of mice using the TIANamp Stool DNA Kits (Tiangen, Beijing, China). Using previously reported methods (18), fecal DNA was amplified by 16S rRNA genes (V3–V4) and sequenced on an Illumina MiSeq Instrument (Illumina, San Diego, CA, USA). The following were the sequences of universal primers (5′–3′): V3-341F (CCTAYGGGRBGCASCAG) and V4-806R (GGACTACNNGGGTATCTAAT). The PCR products were purified using a QIAquick Gel Extraction Kit (Qiagen, Hilden, Germany). Sequencing libraries were generated by instructions for TruSeq DNA PCR-Free Sample Prep Kit (Illumina, San Diego, CA, USA). Sequencing was performed using the Illumina MiSeq platform (Illumina, San Diego, CA, USA) according to the standard protocol (Novogene Co., Ltd., Beijing, China). Briefly, the raw sequencing data were processed by trimming and quality filtering to generate clean reads. The average number of raw sequences per sample was 62625-97171. The detailed sequence counts per sample are provided in Table S1. Denoising was performed using the Divisive Amplicon Denoising Algorithm 2 (DADA2), followed by the removal of low-frequency sequences (counts < 5) to obtain the final amplicon sequence variants (ASVs). Representative sequences of each ASV were taxonomically annotated to characterize phyla, genera, and abundance distribution. Alpha diversity analysis was conducted to assess species richness and evenness within

samples. Additionally, beta diversity and principal coordinate analysis were employed to evaluate differences in microbial community structure across samples or groups. To identify statistically significant variations in phylum and genus composition between sample groups, linear discriminant analysis effect size was applied.

## Validation of bacterial strain entry into the intestine via oral gavage

The 6-week-old BALB/c mice were divided into two groups, five in each group, and were given the same dose of *L. paracasei* 207-27 bacterial solution and normal saline, respectively, for 2 weeks. Fresh mouse feces and extracted bacterial genomic DNA were collected, using *L. paracasei* 207-27 species primers and methods verified by Guo et al. (20), and real-time PCR was performed to determine the abundance level of *Lacticaseibacillus paracasei* in mouse feces (Fig. S5).

## Short-chain fatty acid analysis

Here, 100 mg of mice fecal sample was accurately weighed; homogenized with 100 µL of 15% phosphoric acid, 100 µL of 50 µg/mL internal standard (isocaproic acid), and 400 µL of diethyl ether for 1 min; and subsequently centrifuged for 10 min at 4℃ at 12,000 rpm. The supernatant was carefully transferred to a new 1.5 mL tube for further analysis using the Agilent 7890B gas chromatograph (Agilent, Santa Clara, CA, USA) and Agilent HP-INNOWax capillary column (Agilent, Beijing, China). The following short-chain fatty acids (SCFAs) were included in the assay: acetic, propionic, butyric, isobutyric, valeric, isovaleric, and caproic acids. The following were the chromatographic conditions: split injection, 1 µL injection volume, and 10:1 split ratio. The inlet temperature was set as follows: inlet, 250℃; transfer line, 250℃; ion source, 230℃; and quadrupole, 150℃.

## Secretory immunoglobulin A level assays

Here, 50 mg of cecal contents was homogenized with 200 µL phosphate buffered saline and subsequently centrifuged for 10 min at room temperature at $1,000 \times g$. The supernatant was collected for further analysis. On day 21, the sample supernatant was diluted 2,500 times, and on day 46, it was diluted 10,000 times. The mouse secretory immunoglobulin A (sIgA) was determined using an ELISA Kit (Elabscience Biotechnology Co., Ltd., Wuhan, China) following the manufacturer's instructions.

## Colonic and splenic cytokine mRNA expression

Total RNA was extracted from the colonic and spleen tissues of mice using the Animal Total RNA Isolation Kit (Foregene, Chengdu, China). RNA was reverse transcribed into cDNA using iScript cDNA Synthesis Kit (Bio-Rad, Hercules, CA, USA) in a C1000 Touch Thermal Cycler (Bio-Rad). The following was the reverse transcription-PCR (RT-PCR) reaction process: 25℃, 46℃, and 95℃ for 5, 20, and 1 min, respectively. According to the primer sequences listed in Table 1, real-time RT-PCR (RT-qPCR) of cDNA was performed using the SsoFast EvaGreen Supermix (Bio-Rad) in a CFX96 system (Bio-Rad). The following were the RT-qPCR cycling conditions: reaction with an initial denaturation step at 98℃ for 30 s, followed by 39 cycles of denaturation at 98℃ for 15 s, and annealing temperature with an extension step for 30 s at 60℃. The mRNA expressions of mouse genes were analyzed with the specific primers listed below. The relative mRNA expression was calculated using the comparative cycle method ($2^{-\Delta\Delta Ct}$). β-Actin served as an internal reference gene, the ceftri group as a control group on day 21, and the NS-water group as a control group on day 46.

## Serum cytokine analysis

At the time of sacrifice, the eyeballs of mice were removed to collect the blood samples. The serum was obtained by centrifugation at $2,000 \times g$ for 15–20 min at 4℃. Cytokines, interleukin-5 (IL-5), IL-6, IL-10, IL-13, IL-17A, IL-23, TNF-α, and TGF-β in the

**TABLE 1** Primer sequences for RT-qPCR

| Gene | Primer |
| --- | --- |
| β-Actin-F | 5′-GTGGGCCGCTCTAGGCACCAA-3′ |
| β-Actin-R | 5′-CTCTTTGATGTCACGCACGATTTC-3′ |
| IL-6-F | 5′-GTCACAGAAGGAGTGGCTA-3′ |
| IL-6-R | 5′-AGAGAACAACATAAGTCAGATACC-3′ |
| IL-10-F | 5′-GACCAGCTGGACAACATACT-3′ |
| IL-10-R | 5′-GAGGGTCTTCAGCTTCTCAC-3′ |
| IL-12P40-F | 5′-CTCTGTCTGCAGAGAAGGTC-3′ |
| IL-12P40-R | 5′-GCTGGTGCTGTAGTTCTCAT-3′ |
| TNF-α-F | 5′-CTCTTCAAGGGACAAGGCTG-3′ |
| TNF-α-R | 5′-CGGACTCCGCAAAGTCTAAG-3′ |
| IL-17A-F | 5′-TGATGCTGTTGCTGCTGCTGAG-3′ |
| IL-17A-R | 5′-CACATTCTGGAGGAAGTCCTTGGC-3′ |
| Claudin-1-F | 5′-GCTGGGTTTCATCCTGGCTTCTC-3′ |
| Claudin-1-R | 5′-CCTGAGCGGTCACGATGTTGTC-3′ |
| Occludin-F | 5′-GCGAGGAGCTGGAGGAGGAC-3′ |
| Occludin-R | 5′-CGTCGTCTAGTTCTGCCTGTAAGC-3′ |
| ZO-1-F | 5′-GCGAACAGAAGGAGCGAGAAGAG-3′ |
| ZO-1-R | 5′-GCTTTGCGGGCTGACTGGAG-3′ |
| Ki67-F | 5′-GCCTGCCCGACCCTACAAAATG-3′ |
| Ki67-R | 5′-CTCATCTGCTGCTGCTTCTCCTTC-3′ |
| MUC2-F | 5′-TGCTGACGAGTGGTTGGTGAATG-3′ |
| MUC2-R | 5′-TGATGAGGTGGCAGACAGGAGAC-3′ |

serum were determined using ELISA kits (R&D Systems, Minneapolis, MN, USA) following the manufacturer's instructions, respectively. Finally, the levels of these cytokines in the serum of mice were measured using the Luminex assay (R&D Systems) using a Luminex 200 multiplexing instrument (Merck Millipore).

## Statistical analysis

Statistical analysis was performed using Statistical Package for the Social Sciences (version 25, IBM, Armonk, NY, USA). Data were presented as means ± standard deviations. One-way analysis of variance or Kruskal–Wallis H-test was used for multiple comparisons. $P < 0.05$ was considered statistically significant.

## RESULTS

### Protective effect of DSS-induced colitis at day 46

To examine the protective effect of *L. paracasei* 207-27, the histopathological damage was evaluated using H&E staining. Compared with those in the NS-water group, inflammatory scores in the NS–DSS group were markedly increased (NS-water group, 2.10 ± 1.62; NS–DSS group, 11.10 ± 2.63) ($P < 0.0001$). Histologically, the NS-water group showed intact colonic epithelial cells and colonic mucosa, neat intestinal villi, abundant goblet cells, and inflammatory cells that were not infiltrated (Fig. 1B), whereas the NS–DSS group showed incomplete mucosal structures, crypt abscesses, ulcers, and extensive inflammatory cell infiltration in the colonic tissues (Fig. 1B), indicating successful colitis modeling. As shown in Fig. 1A, the 3-week *L. paracasei* 207-27 treatment in early life significantly protected the DSS-induced symptoms of colitis with lower inflammatory scores (207-27 short-term group, 9.30 ± 2.76; ceftri–DSS group, 10.20 ± 3.22) ($P < 0.05$). The colonic tissues in the ceftri–DSS group showed similar damage to the NS–DSS group, whereas the 207-27 short-term group showed less inflammatory cell infiltration and fewer ulcers. However, the long-term *L. paracasei* 207-27 application did not significantly reduce inflammation scores (207-27 long-term group, 10.70 ± 2.75).

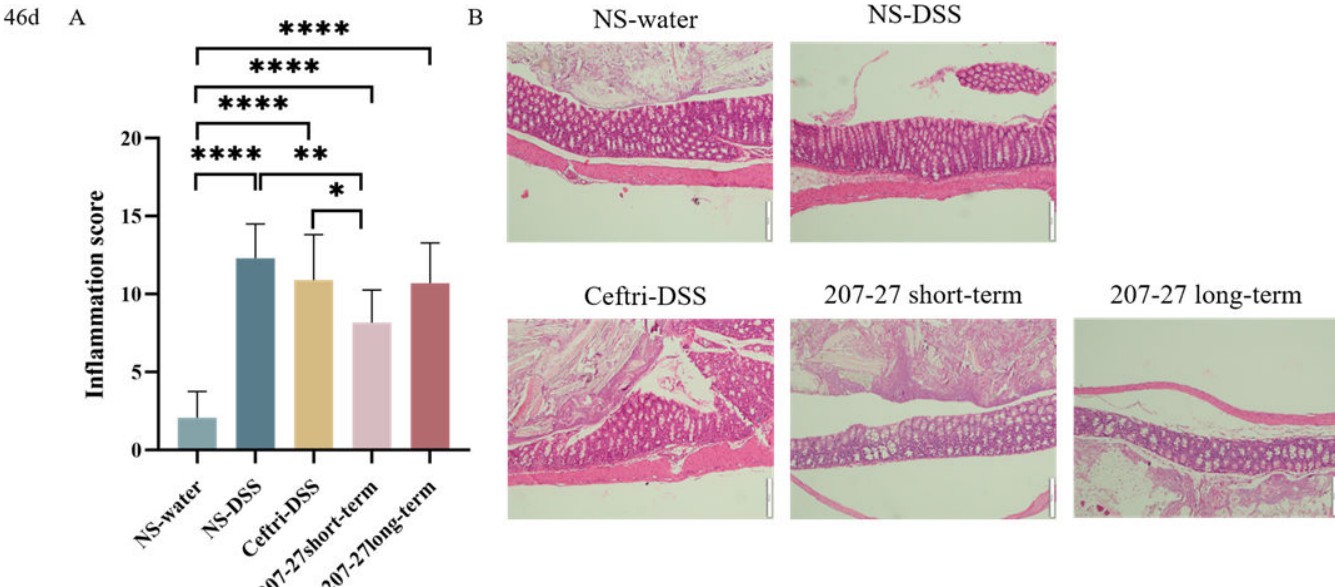

**FIG 1** Colonic inflammation following dextran sulfate sodium administration (46 days, NS-water group, $n = 14$; NS–DSS group, $n = 10$; ceftri–DSS group, $n = 10$; 207-27 short-term group, $n= 11$; and 207-27 long-term group, $n = 10$). (A) Inflammatory pathology score (mean ± standard deviation). (B) Typical histological images of H&E-stained colonic tissues in the NS-water, NS–DSS, ceftri–DSS, 207-27 short-term, and 207-27 long-term groups. Microscopic magnification: 10×. *$P < 0.05$, **$P < 0.01$, ***$P < 0.001$, and ****$P < 0.0001$ as conducted.

## Changes in the gut microbiota and metabolites after treatment on day 21

On day 21, after continuous gavage of 207-27, gut microbiota composition was determined using 16S rRNA gene high-throughput sequencing. At the phylum level (Fig. 2A), the relative abundance of *Bacteroidota* and *Verrucomicrobiota* was the highest in the NS group (0.40 and 0.32, respectively). The ceftri group had the highest relative abundance of *Firmicutes* (ceftri group, 0.96; NS group, 0.24; and ceftri + 207-27 group, 0.93). At the genus level (Fig. 2B), the ceftri group had a higher relative abundance of *Ligilactobacillus* (0.66) and *Enterococcus* (0.17), and a lower relative abundance of *Akkermansia* (0.005) than the NS group ($P < 0.05$, $P < 0.05$, and $P < 0.0001$, respectively). The ceftri + 207-27 group had a higher relative abundance of *Lacticaseibacillus* (0.11) than the NS and ceftri groups (both, $P < 0.05$). The relative abundance of *Enterococcus* had no statistical difference between the ceftri + 207-27 and ceftri groups ($P > 0.05$). *Clostridium* was not detected in any of the groups. It has been confirmed that the bacterial strain reached the intestine via oral gavage and can be detected in fecal samples, indicating its ability to tolerate the digestive process and potentially colonize the gut to modulate the gut microbiota, as shown in Fig. S5.

Regarding the alpha diversity of fecal microbiota (Fig. 2C), the ceftri group had significantly lower ACE, Chao1, Shannon, and Simpson indexes than the NS group ($P < 0.001$, $P < 0.001$, $P < 0.001$, and $P < 0.05$, respectively), revealing both the richness and diversity of the gut microbiota composition. The 207-27 group had higher ACE, Chao1, Shannon, and Simpson indexes than the ceftri group ($P < 0.01$, $P < 0.01$, $P < 0.001$, and $P < 0.05$, respectively).

Compared with the NS group, acetic, propionic, butyric, isobutyric, valeric, and isovaleric acid levels were decreased in the ceftri group ($P < 0.01$, $P < 0.0001$, $P < 0.05$, $P < 0.001$, $P < 0.01$, and $P < 0.01$, respectively) (Fig. 2D and Fig. S3A). However, compared with the ceftri group, the 207-27 treatment increased the acetic, valeric, and hexanoic acid levels ($P < 0.05$, $P < 0.05$, and $P < 0.001$, respectively).

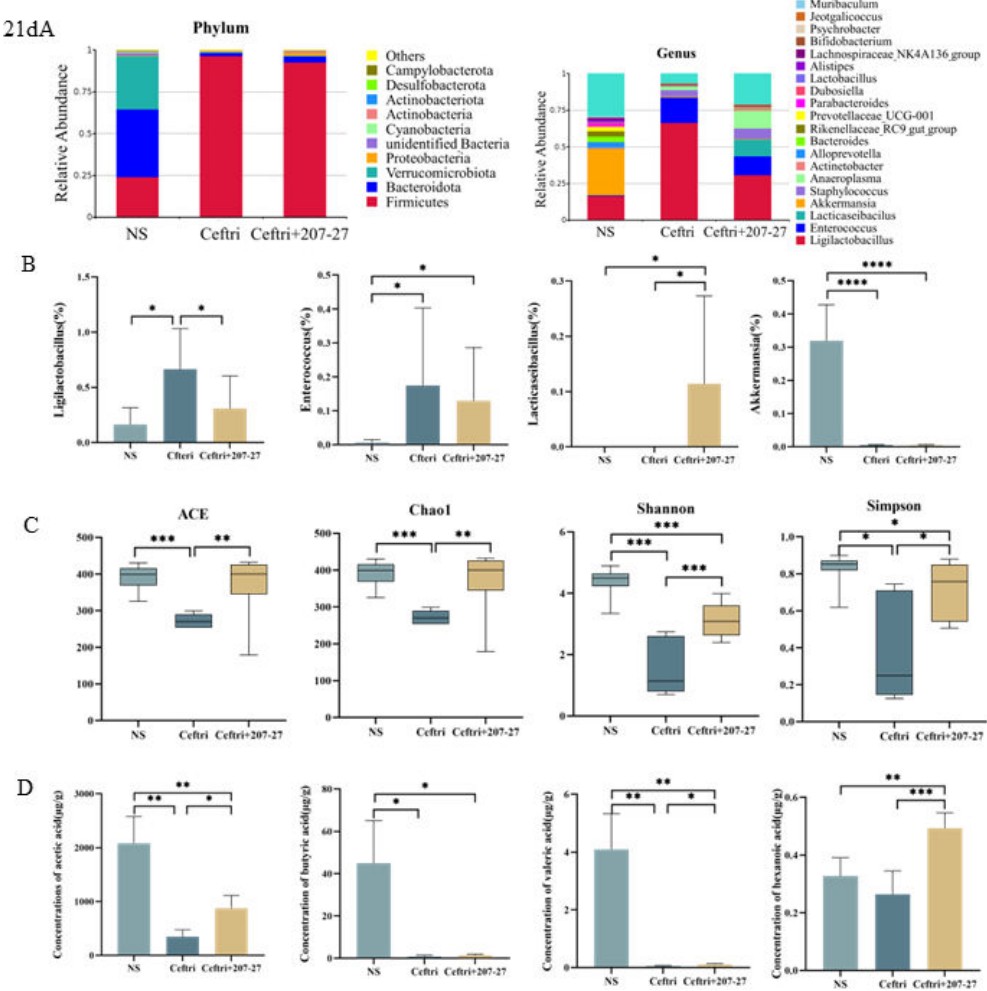

**FIG 2** Effects on microbiota and metabolites on day 21. (A–C) (NS group, $n = 12$; ceftri group, $n = 6$; and ceftri + 207-27 group, $n = 12$) and (D) $n = 5$. (A) Relative abundance at the phylum and genus levels. (B) Relative abundance of *Ligilactobacillus*, *Enterococcus*, *Lacticaseibacillus*, and *Akkermansia*. (C) The alpha diversity of the gut microbiota. (D) SCFA concentrations (acetic, butyric, valeric, and hexanoic acids). *$P < 0.05$, **$P < 0.01$, ***$P < 0.001$, and ****$P < 0.0001$ as conducted.

## Intestinal development after treatment on day 21

To investigate the immediate effect of 207-27 on the colon epithelium barrier after 21 days of early-life use, we detected the mRNA expression levels of intestinal epithelial cells and tight junction proteins, including Ki67, occludin, claudin, MUC2, and ZO-1. Compared with the NS group, ceftriaxone administration decreased the Ki67 and occludin mRNA levels (both, $P < 0.01$) (Fig. 3A and B). The ceftri + 207-27 group showed an upward trend in the Ki67 and occludin mRNA levels compared with the ceftri group; however, the difference was not statistically significant (both, $P > 0.05$) (Fig. 3A and B). The ceftri + 207-27 group showed lower claudin and MUC2 mRNA levels than the NS group (both, $P < 0.01$), and no significant difference was noted between the claudin and MUC2 mRNA levels in the ceftri and NS groups (both, $P > 0.05$) (Fig. 3C and D). Meanwhile, ZO-1 mRNA levels had no difference between the groups (all, $P > 0.05$) (Fig. 3E).

sIgA plays a key role in mucosal immune function and is a significant substance in maintaining intestinal mucosal homeostasis. The ceftri and ceftri + 207-27 groups had significantly higher sIgA levels than the NS group ($P < 0.01$ and $P < 0.001$, respectively) (Fig. 3F).

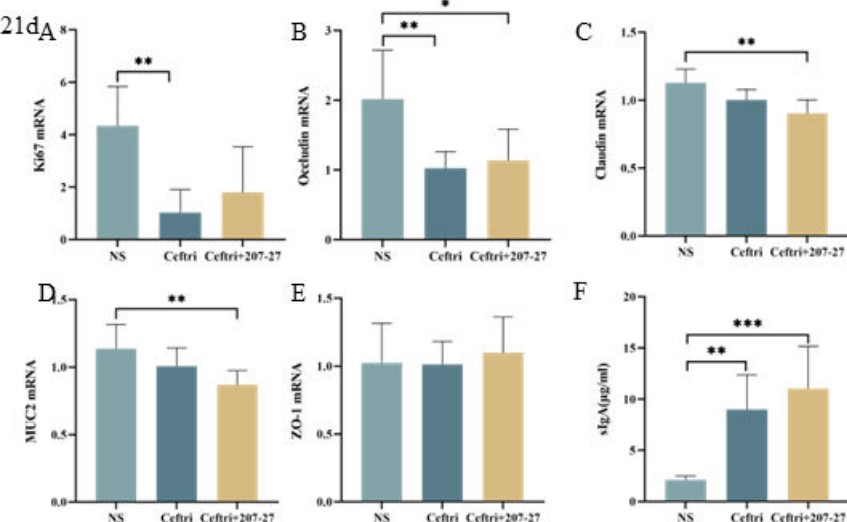

**FIG 3** Colonic mucosal barrier after intervention on day 21 ($n$ = 4–7). (A) Colonic Ki67 mRNA level. (B) Colonic occludin mRNA level. (C) Colonic claudin mRNA level. (D) Colonic MUC2 mRNA level. (E) Colonic ZO-1 mRNA level. (F) sIgA level in the cecum feces. *$P < 0.05$, **$P < 0.01$, and ***$P < 0.001$ as conducted.

## Colonic and systemic immune response on day 21

To understand the reason why short-term *L. paracasei* 207-27 administration could relieve colitis, we assessed typical pro-inflammatory and anti-inflammatory cytokine mRNA levels after 21 days of continuous gavage of 207-27. In the colon homogenate supernatant, the ceftri group showed higher IL-12 mRNA levels than the NS group ($P < 0.05$) (Fig. S2C). The 207-27 group showed lower IL-12 mRNA levels than the ceftri group ($P < 0.05$) (Fig. S2C). The IL-10 and TGF-β mRNA levels in the 207-27 group showed an increasing trend compared with those in the ceftri group (both, $P > 0.05$) (Fig. 4A and B). In the spleen homogenate supernatant, the ceftri group had lower IL-17 mRNA levels than the NS group ($P < 0.05$) (Fig. 4K). The mRNA levels of anti-inflammatory cytokines including IL-10, IL-13, and TGF-β and pro-inflammatory cytokines including IL-5, IL-6, and IL-17 were increased in the ceftri + 207-27 group compared with those in the ceftri group ($P < 0.01$, $P < 0.05$, $P < 0.01$, $P < 0.05$, $P < 0.05$, and $P < 0.01$, respectively) (Fig. 4E through K). In serum, the IL-5, IL-6, and TNF-α mRNA levels in the ceftri group showed a decreasing trend compared with those in the NS group (all, $P > 0.05$) (Fig. 4M, N, and P). The ceftri + 207-27 group showed a significantly decreased IL-6 mRNA level compared with the ceftri group ($P < 0.05$) (Fig. 4N).

## Changes in the gut microbiota and metabolites after treatment on day 46

To examine the effect of 207-27 continuously used under early-life antibiotic exposure, we analyzed the gut microbiota composition on day 46. At the phylum level (Fig. 5A), the main species detected in the NS-water group were *Firmicutes*, *Bacteroidota*, and *Verrucomicrobiota* (0.47, 0.37, and 0.11, respectively). The NS–DSS group had fewer *Bacteroidota* (0.24) and more *Verrucomicrobiota* (0.16), *Proteobacteria* (0.05), and *Euryarchaeota* (0.03) than the NS-water group. The ceftri–DSS group had the highest *Verrucomicrobiota* (0.45), the lowest *Firmicutes* (0.27), and *Bacteroidota* (0.17). The 207-27 long-term group had a higher relative abundance of *Firmicutes* than the ceftri–DSS and 207-27 short-term groups ($P < 0.01$ and $P < 0.05$, respectively). At the genus level (Fig. 5A and B), the NS–DSS group had lower *Lacticaseibacillus* than the NS-water group ($P < 0.05$). Compared with the NS–DSS group, the ceftri–DSS group had a higher relative abundance of *Akkermansia* ($P < 0.05$) and had a downward trend in the relative abundance of *Ligilactobacillus*, *Dubosiella,* and *Lactobacillus* (all, $P > 0.05$). Regarding the

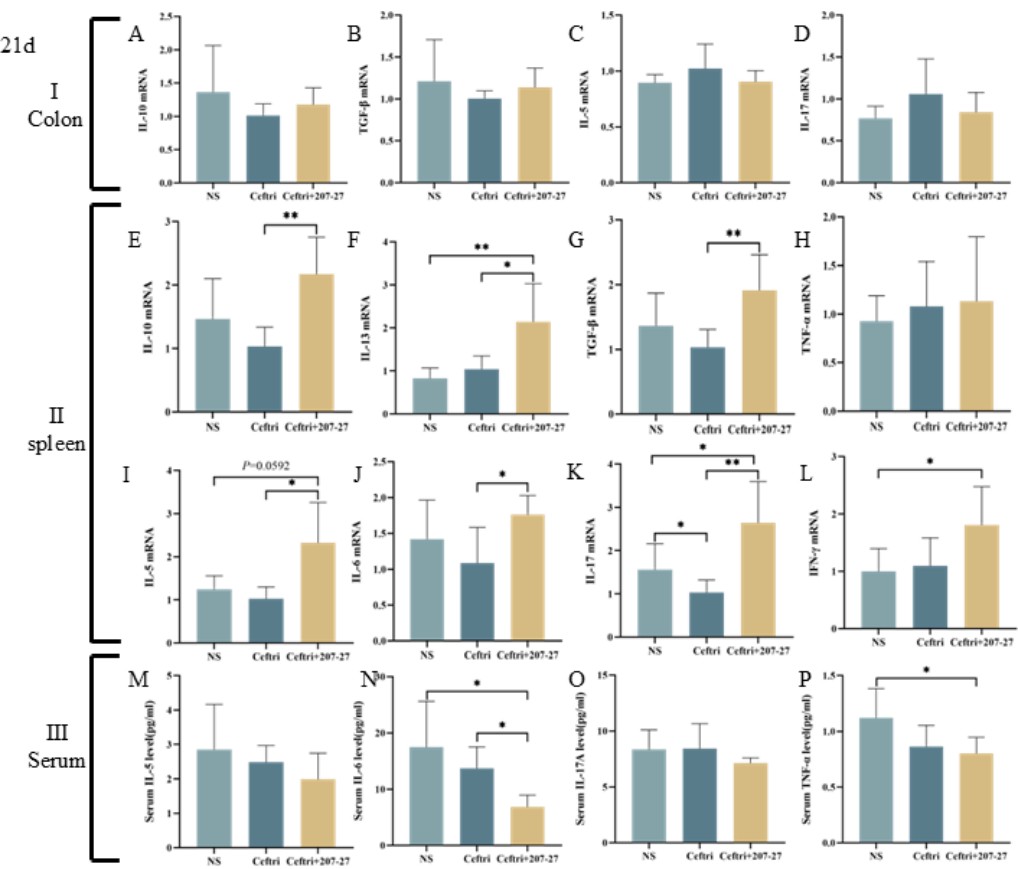

**FIG 4** Local and systemic immunities after intervention on day 21 (*n* = 5–8). (A) Colonic IL-10 mRNA level. (B) Colonic TGF-β mRNA level. (C) Colonic IL-5 mRNA level. (D) Colonic IL-17 mRNA level. (E) Splenic IL-10 mRNA level. (F) Splenic IL-13 mRNA level. (G) Splenic TGF-β mRNA level. (H) Splenic TNF-α mRNA level. (I) Splenic IL-5 mRNA level. (J) Splenic IL-6 mRNA level. (K) Splenic IL-17 mRNA level. (L) Splenic IFN-γ mRNA level. (M) Serum IL-5 level. (N) Serum IL-6 level. (O) Serum IL-17A level. (P) Serum TNF-α level. *$P < 0.05$ and **$P < 0.01$ as conducted.

relative abundance of *Lactobacillus*, the 207-27 short-term group showed an increasing trend compared with the ceftri–DSS group ($P = 0.0777$), and the 207-27 long-term group was significantly higher than the ceftri–DSS and 207-27 short-term groups (both, $P <$ 0.01). No significant difference in the relative abundance of *Enterococcus* was noted in the groups, and *Clostridium* was not detected.

Regarding the alpha diversity of fecal microbiota (Fig. 5C), the ceftri–DSS group had significantly lower Shannon and Simpson indexes than the NS-water group (both, $P <$ 0.05). Furthermore, the ceftri–DSS had a lower Simpson index than the NS–DSS group ($P < 0.05$). The 207-27 long-term group had a significantly higher Simpson index than the ceftri–DSS group ($P < 0.05$), indicating both the richness and diversity of the gut microbiota composition.

At the SCFA level (Fig. S3), increasing trends in the acetic, isobutyric, valeric, and isovaleric acid concentrations were noted in the NS–DSS group compared with those in the NS-water group (all, $P > 0.05$). The ceftri–DSS group had higher butyric and isobutyric acid concentrations than the NS–DSS group (both, $P < 0.05$). The 207-27 long-term group had significantly higher butyric and valeric acid concentrations than the 207-27 short-term group (both, $P < 0.05$).

## Intestinal development after treatment on day 46

As shown in Fig. 6, no significant difference is observed between the NS-water and NS–DSS groups in the Ki67, occludin, claudin, MUC2, and ZO-1 mRNA levels ($P > 0.05$)

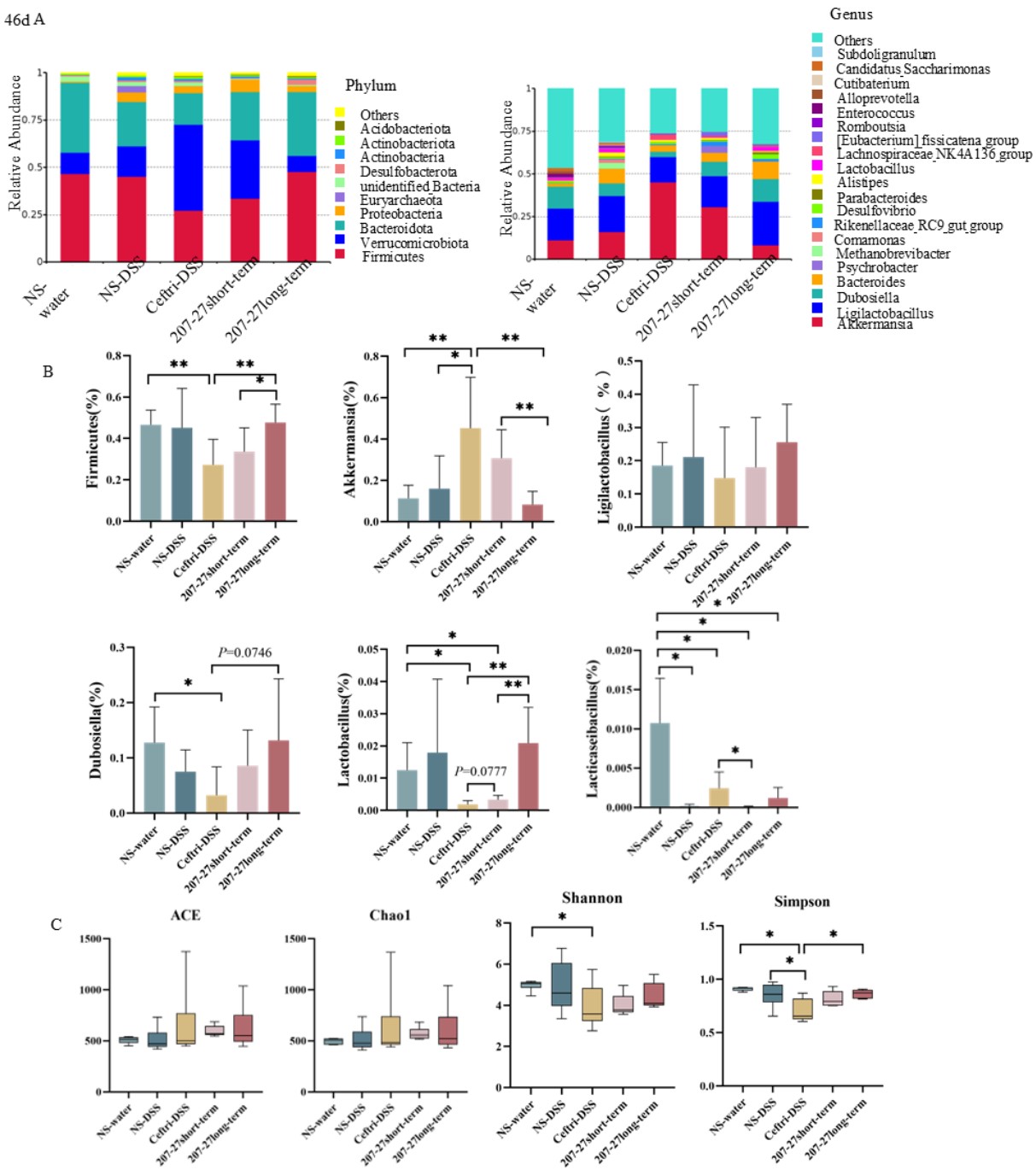

**FIG 5** Effects on microbiota and metabolites on day 46 (*n* = 6). (A) Relative abundance at the phylum and genus levels. (B) Relative abundance of *Firmicutes*, *Akkermansia*, *Ligilactobacillus*, *Dubosiella*, *Lactobacillus*, and *Lacticaseibacillus*. (C) Alpha diversity of the gut microbiota. *P < 0.05 and **P < 0.01 as conducted.

(Fig. 6A through E). The ceftri–DSS group had lower occludin mRNA levels than the NS-water group (*P* < 0.01) (Fig. 6B). However, the ceftri–DSS group had higher sIgA levels than the NS-water group (*P* < 0.01) (Fig. 6F). The 207-27 long-term group had higher claudin mRNA level and sIgA level than the ceftri–DSS group (*P* < 0.05 and *P* < 0.0001, respectively) (Fig. 6C and F). The 207-27 long-term group had higher claudin mRNA level, MUC2 mRNA level, and sIgA level than the 207-27 short-term group (*P* < 0.05, *P* < 0.05, and *P* < 0.0001, respectively) (Fig. 6C, D, and F).

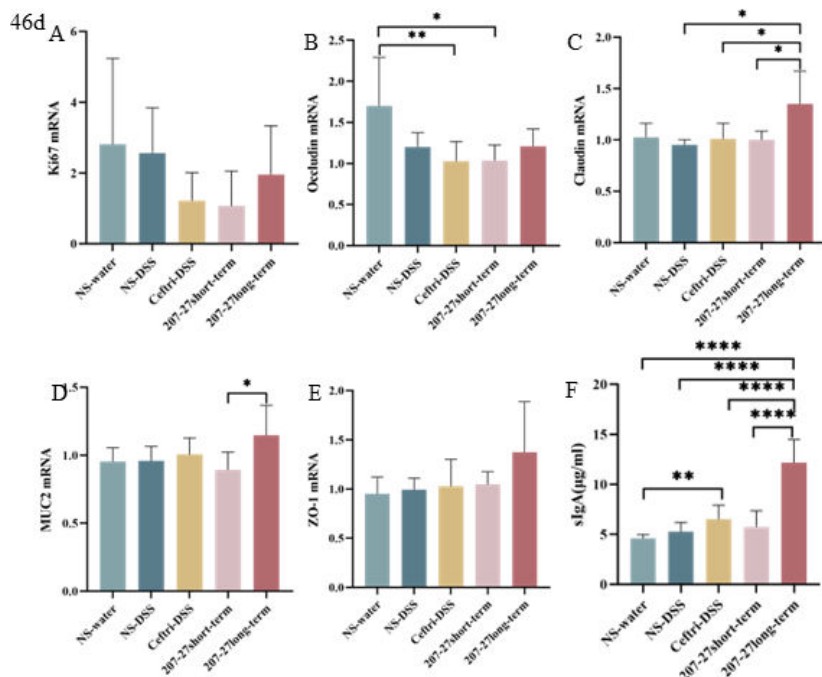

**FIG 6** Colonic mucosal barrier after intervention on day 46 (*n* = 5–8). (A) Colonic Ki67 mRNA level. (B) Colonic MUC2 mRNA level. (C) sIgA level in the cecum feces. (D) Colonic ZO-1 mRNA level. (E) Colonic claudin mRNA level. (F) Colonic occludin mRNA level. *P* < 0.05, **P* < 0.01, ***P* < 0.001, and ****P* < 0.0001 as conducted.

## Colonic and systemic immune response on day 46

The expression levels of colonic, splenic, and serum immune factors on day 46 are demonstrated in Fig. 7 and Fig. S4. No significant difference was observed between the NS-water and NS–DSS groups in terms of colonic pro-inflammatory and anti-inflammatory cytokines (all *P* > 0.05). The 207-27 short-term and 207-27 long-term groups had significantly reduced colonic IFN-γ mRNA levels compared with the ceftri–DSS group (*P* < 0.05 and *P* < 0.001, respectively) (Fig. 7C). The 207-27 short-term group showed a decreasing trend in colonic IL-17 levels compared with the ceftri–DSS group (*P* = 0.0556) (Fig. 7D). The 207-27 long-term group had a significant decrease in colonic TNF-α levels compared with the ceftri–DSS group (*P* < 0.05) (Fig. 7E). In contrast, the 207-27 long-term group showed an increasing trend in colonic anti-inflammatory TGF-β mRNA levels compared with the ceftri–DSS group (*P* = 0.0506) (Fig. 7A). In the spleen homogenate supernatant, compared with the NS-water group, the mRNA levels of pro-inflammatory cytokines such as IL-5 and IL-17 and anti-inflammatory cytokines such as IL-10 and IL-13 were decreased in the NS–DSS group (*P* < 0.001, *P* < 0.05, *P* < 0.01, and *P* < 0.0001, respectively) (Fig. 7G and I, and Fig. S4E and F). The ceftri–DSS group had higher pro-inflammatory IL-12 mRNA levels than the NS-water and NS–DSS groups (*P* < 0.01 and *P* < 0.01, respectively) (Fig. S4G). The 207-27 short-term group had decreased pro-inflammatory IL-5, IL-6, and TNF-α mRNA levels compared with the ceftri–DSS group (*P* < 0.05, *P* < 0.01, and *P* < 0.05, respectively); however, the 207-27 long-term group had significantly higher TNF-α levels than the 207-27 short-term group (*P* < 0.01). In serum, the NS–DSS group had lower TGF-β mRNA levels and higher IL-5 mRNA levels than the NS-water group (*P* < 0.001 and *P* < 0.05, respectively) (Fig. 7K and L). The ceftri–DSS group had significantly lower TGF-β levels and higher pro-inflammatory IL-6, IL-17A, and TNF-α levels than the NS-water group (*P* < 0.001, *P* < 0.05, *P* < 0.05, and *P* < 0.05, respectively) (Fig. 7K, M through O). The 207-27 long-term group had lower IL-6 and TNF-α levels than the ceftri–DSS group (*P* < 0.0001 and *P* < 0.001, respectively) (Fig.

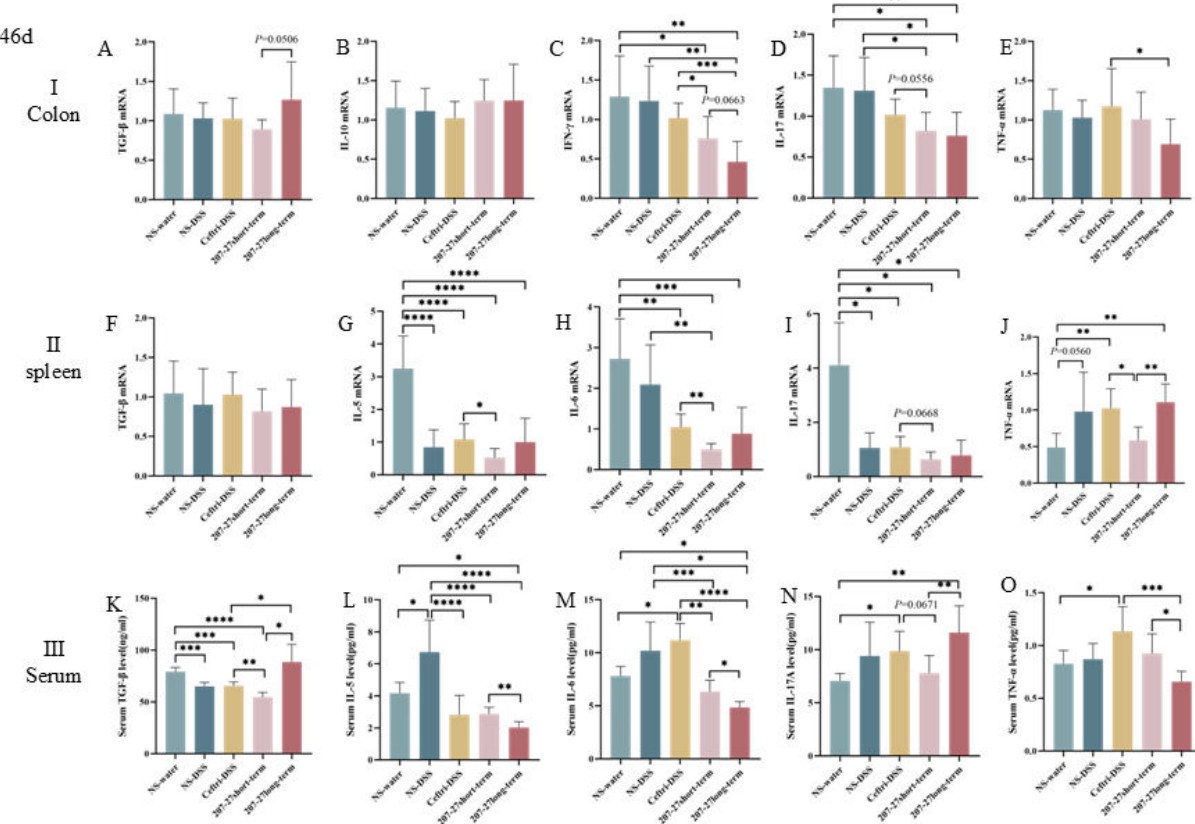

**FIG 7** Local and systemic immunities after intervention on day 46 (*n* = 5–8). (A) Colonic TGF-β mRNA level. (B) Colonic IL-10 mRNA level. (C) Colonic IFN-γ mRNA level. (D) Colonic IL-17 mRNA level. (E) Colonic TNF-α mRNA level. (F) Splenic TGF-β mRNA level. (G) Splenic IL-5 mRNA level. (H) Splenic IL-6 mRNA level. (I) Splenic IL-17 mRNA level. (J) Splenic TNF-αmRNA level. (K) Serum TGF-β level. (L) Serum IL-5 level. (M) Serum IL-6 level. (N) Serum IL-17A level. (O) Serum TNF-α level. *$P <$ 0.05, **$P <$ 0.01, ***$P <$ 0.001, and ****$P <$ 0.0001 as conducted.

7M and O). The 207-27 long-term group had significantly lower serum pro-inflammatory IL-5, IL-6, and TNF-α levels than the 207-27 short-term group ($P <$ 0.01, $P <$ 0.05, and $P <$ 0.05, respectively) (Fig. 7L, M, and O). The 207-27 long-term group had significantly higher serum TGF-β and IL-17A levels than the 207-27 short-term group ($P <$ 0.05 and $P <$ 0.01, respectively) (Fig. 7K and N).

## DISCUSSION

Early-life antibiotic treatments influence a large fraction of the global population and are associated with global epidemic health problems; by supplementing probiotics, correcting undesired changes in the gut microbiota composition and function caused by antibiotic treatments is possible (14). Early life is a significant period of susceptibility for IBD development later in life (15). Accumulating evidence has indicated that probiotics can alleviate symptoms of DSS-induced colitis in a strain-specific way (7–9). Previous studies reported that *L. paracasei* 207-27 has potential anti-inflammatory and anti-allergic effects (10–12). However, few studies have focused on whether *L. paracasei* 207-27 restores the intestinal barrier damage caused by antibiotics, and whether *L. paracasei* 207-27 also has a favorable role in DSS-induced colitis. We here demonstrated that short-term *L. paracasei* 207-27 strain administration regulates the immune response and improves the symptoms of DSS-induced colitis via the gut microbiota, and we observed that long-term *L. paracasei* 207-27 administration enhanced the intestinal barrier.

Intestinal inflammation scoring and pathologic H&E sections assessed colitis severity and colonic structural changes. The results showed that the NS–DSS group scored

higher than the NS-water group and had incomplete mucosal structures, crypt abscesses, ulcers, and extensive inflammatory cell infiltration, indicating successful colitis modeling (21). Similarly, the early-life antibiotic exposure group showed colitis symptoms. Several studies have suggested that early-life perturbation can increase colitis risk later in life (22, 23). Following early-life antibiotic exposure, mice were treated short- and long-term with *L. paracasei* 207-27, and the short-term 207-27 group scored lower than the ceftri–DSS and long-term 207-27 groups. The results suggest that short-term *L. paracasei* 207-27 administration in mice can restore antibiotic damage, attenuate the symptoms of future DSS-induced colitis, protect the intestinal structure, and reduce the inflammatory response. However, the continued 6-week *L. paracasei* 207-27 administration for 6 weeks did not significantly reduce intestinal inflammation scores. This may be because inflammation relief and intestinal flora adjustment require only short-term probiotic colonization, whereas restoration of the intestinal barrier requires long-term sustained probiotic colonization. To investigate the reasons for the different effects of short-term and long-term *L. paracasei* 207-27 administration, we further analyzed the possible pathways and mechanisms by which *L. paracasei* 207-27 exerts its protective effects in terms of gut microbiota and metabolites, intestinal development, and immune responses.

Several studies have shown that early-life antibiotic exposure is associated with a disturbance in the gut microbiota and a reduction in alpha diversity (14, 23, 24). In our study, antibiotic administration in early life decreased the alpha diversity index, increased the relative abundance of *Enterococcus*, and decreased the relative abundance of *Akkermansia*. Short-term *L. paracasei* 207-27 administration increased the alpha diversity index of the gut microbiota, increased the relative abundance of *Lacticaseibacillus*, and decreased the relative abundance of *Ligilactobacillus* and *Enterococcus*. The results showed that short-term *L. paracasei* 207-27 administration could restore unbalanced gut microbiota caused by antibiotics and increase the level of potential beneficial bacteria. Carpay et al. reported that prebiotic, probiotic, or synbiotic supplementation in infants who received antibiotics early in life mostly increased the phyla, families, genera, and species that corresponded to the probiotic or synbiotic intervention administered (25), which is confirmed in the present study. *Ligilactobacillus* is a promising lactic acid bacterium with anti-inflammatory effects and can maintain gut microbiota balance (26). In this study, on day 21, *Ligilactobacillus* decreased following *L. paracasei* 207-27 intervention, probably because of a competitive effect with *L. paracasei* 207-27. Following long-term *L. paracasei* 207-27 administration, the decline in the Simpson index caused by the early-life antibiotic intervention recovered. However, other alpha diversity indexes, including ACE, Chao, and Shannon, did not significantly improve. Ozkul et al. suggested that long-term antibiotic administration can decrease the dominant microbial communities and an overgrowth of foreign or transient microbial communities, which may lead to a further increase in alpha diversity (22). We modeled colitis using DSS to investigate the effect of *L. paracasei* 207-27 administration following early-life antibiotic exposure on future colitis. The ceftri–DSS group without probiotic intervention decreased gut microbiota diversity, increased the relative abundance of *Akkermansia*, and decreased the relative abundance of the beneficial genus *Lacticaseibacillus*. *Akkermansia* is a gram-negative anaerobic bacterium that is selectively decreased in the fecal microbiome of patients with IBD (27). However, in our study, the ceftri–DSS group did not reduce the relative abundance of *Akkermansia*, probably because antibiotics reduced the gut microbiota diversity and instead promoted *Akkermansia* growth. Moreover, on day 46, no increase in *Akkermansia* was noted in the 207-27 long-term group, which may be related to the increase in other beneficial bacteria, such as *Lactobacillus*, *Dubosiella*, and *Ligilactobacillus*, or it may also be because *L. paracasei* 207-27 has an anti-gram-negative effect, and the long-term application affects the *Akkermansia* instead (10). Long-term *L. paracasei* 207-27 administration restored the gut microbiota diversity. Emerging evidence suggests that gut microbiota dysbiosis contributes to IBD pathogenesis (28). Our study observed that *L. paracasei*

207-27 protects against colitis by restoring the gut microbiota diversity. Interestingly, despite additional short-term or long-term *L. paracasei* 207-27 administration, the relative abundance of *Lacticaseibacillus* did not significantly increase. Figure 5B shows the microbiota data at the end of the experiment, in which the NS-water and NS–DSS groups were not given antibiotics and the strains were not given by gavage, so the measured *Lacticaseibacillus* levels are the natural level in the mice, which are really not very high in terms of relative abundance and number (0.01%–0.02%). The 207-27 short-term group was gavaged with both antibiotics and bacteria for 3 weeks. Because the antibiotics killed the original *Lacticaseibacillus*, the level of *Lacticaseibacillus* was lower in this group after discontinuing the gavage and was slightly higher in this group compared to the control group (ceftri–DSS), which was also treated with antibiotics, but there was no difference in *P* value. It is likely because of the priority effects of established gut microbiota, which has already undergone a selection process to establish community stability and resilience (29). Under these conditions, the "foreign" *L. paracasei* 207-27 is also excluded from the community. However, the 207-27 long-term group continued to gavage *L. paracasei* 207-27 up to 6 weeks after discontinuing antibiotics on 3 weeks, so the abundance of *Lactobacillus* remained high at the end of the experiment. In addition, as can be seen in Fig. 2B, when the antibiotic and *L. paracasei* 207-27 were administered by gavage together for 3 weeks, the abundance of *Lacticaseibacillus* actually increased compared to the antibiotic group, which suggests that the *L. paracasei* 207-27 that we gavaged entered the intestine after the antibiotics had killed the intestinal native bacteria. Our previous study showed that *L. paracasei* 207-27 modulated the gut microbiota composition in a strain-dependent manner, thereby increasing health-promoting taxa and SCFA levels, particularly butyric acid (10). SCFAs, including butyrate, exert immunomodulatory and anti-inflammatory effects in IBD (30). Our study noted that early-life *L. paracasei* 207-27 intervention restored the ceftriaxone-induced decrease in SCFA levels, including acetic, valeric, and hexanoic acids. Furthermore, *L. paracasei* 207-27 long-term intervention was associated with an increase in the relative abundance of SCFA-producing bacteria, including *Lactobacillus* and *Dubosiella*, and a significant increase in fecal butyric acid compared with short-term intervention. This finding is likely one of the reasons for the different inflammatory pathology scores between the two groups. In this experiment, due to the limitations of the animal experimental design, the phenomenon of changes in the intestinal flora of mice may be affected by the systematic errors brought about by the feeding cage effect and the neonatal mouse litter effect, and more evidence from population-based experiments is needed for the future application of this strain.

Early-life antibiotic exposure impacts immune system development and disrupts gut barrier function, thereby increasing the risk of chronic colitis (15). Ki67 is believed to reflect colonic epithelial cell proliferation (31). MUC2 is believed to reflect the function of the gut barrier and protect epithelia from harmful factors (32). Occludin, ZO-1, and claudin are significant transmembrane and intercellular tight junctions (18, 31). Our study showed that early-life antibiotic exposure disrupted the gut barrier structure, thereby reducing Ki67 and occludin expressions. Cheng et al. noted that ceftriaxone significantly altered the morphology and function of the intestinal epithelium in neonatal mice (33). Furthermore, Shi et al. observed that the level of tight junction protein occludin in the colon was decreased to 50% following antibiotic administration (34). Following short-term *L. paracasei* 207-27 administration, Ki67 and occludin levels slightly recovered, which is consistent with the study on *Bifidobacterium longum* CCM7952 (35) and *Bifidobacterium infantis* 79 (36). However, no significant change in other mucosa and tight junctions was observed, probably because fewer cells led to difficult tight junction detection. sIgA is the most significant antibody in intestinal mucosal immunity, which enhances the mucosal barrier function and simultaneously maintains immune homeostasis (37). Following early-life antibiotic exposure, a specific increase in sIgA levels is noted, which may be caused by changes in the gut microbiota, with specific pathogenic bacteria stimulating the compensatory production of sIgA. The

greater elevation of sIgA after short-term *L. paracasei* 207-27 administration may result from the ability to promote sIgA secretion even after inhibiting pathogenic bacterial colonization. Several probiotics, including *Lactobacillus plantarum* and *L. paracasei,* can promote sIgA production and thus modulate the immune status (38, 39). Similarly, in our previous study, *L. paracasei* 207-27 significantly increased cecal sIgA levels in allergic mice (11). Furthermore, regarding gut immunity, no apparent change in pro-inflammatory and anti-inflammatory cytokines was observed. However, regarding systemic immunity, an upward trend in anti-inflammatory cytokines and a downward trend in pro-inflammatory cytokines were noted. We hypothesized that early-life changes in gut microbiota due to *L. paracasei* 207-27 administration can affect mucosal immunity and, in turn, affect systemic immunity.

Probiotic supplementation can enhance gut barrier function and regulate various inflammation factors, alleviating the symptoms of DSS-induced colitis (9, 28, 35, 40, 41). In the present study, the short-term *L. paracasei* 207-27 administration did not significantly protect the gut barrier, whereas the long-term *L. paracasei* 207-27 administration improved claudin and MUC2, with a tendency to promote intestinal barrier recovery. Moreover, we observed that the levels of splenic anti-inflammatory factors IL-10 and IL-13 decreased, and the serum pro-inflammatory factors IL-6, IL-17A, and TNF-α increased in response to DSS stimulation and early-life antibiotic exposure. INF-γ is one of the most severely upregulated cytokines in IBD and related mouse models (42–44). In this study, long-term administration of *L. paracasei* 207-27 significantly reduced colonic INF-γ expression in the DSS-induced colitis and was more effective than short-term administration of *L. paracasei* 207-27. Short-term and long-term *L. paracasei* 207-27 administration had an anti-inflammatory tendency, which is similar to *Bifidobacterium pseudocatenulatum* MY40C (35), *B. pseudocatenulatum* CCFM680 (35), *B. longum* CCFM681 (40), and *Lactobacillus casei* Zhang (41). Studies have demonstrated that the utilization of probiotics could relieve colitis symptoms via the regulation of various inflammation factors (9, 28, 41). Therefore, we hypothesized that although short-term *L. paracasei* 207-27 administration did not promote gut mechanical barrier, a long-term immune protective effect may exist. Three weeks after stopping antibiotic exposure, gut barrier promotion by the short-term *L. paracasei* 207-27 administration was diminished; however, its effects on systemic immunity remained sustained, thereby playing a protective effect against colitis. Continuous *L. paracasei* 207-27 administration perpetuated the promotion of gut barrier but showed no anti-colitis function, which is likely because the continued use of single-strain stimulation after antibiotic exposure is unhelpful in gut microbiota diversity restoration; therefore, it was inferior to the 207-27 short-term group in systemic immunity, such as splenic IL-5, IL-6, and serum IL-17A levels. Early-life antibiotic exposure may affect the homeostasis and function of the neonatal gut microbiota (45), and probiotic administration does not seem to restore the gut microbiota composition and diversity but rather produces beneficial effects by promoting the increase in other beneficial bacteria (46). Owing to the possible complementary effects of multiple strains, we noted that probiotic mixes have shown better improvements in gut health than single strains (47). Moreover, probiotic administration following early-life antibiotic exposure is significant and should be considered when developing prevention strategies. A clear evidence on when to use probiotics is lacking (48, 49). Similar to the study by Zhong et al. (46), our previous study showed that *L. paracasei* 207-27 modulated the gut microbiota composition in a strain-dependent manner, thereby decreasing gram-negative bacteria and increasing health-promoting taxa and SCFA levels, particularly butyric acid (10). To determine the anti-colitis function of *L. paracasei* 207-27, further studies are needed.

Peng et al. found that early intervention with BD-1 and BD-1 + ceftri had a protective and preventive effect in a DSS-induced colitis model, as evidenced by a reduction in inflammation scores and myeloperoxidase activity levels (18). This is consistent with the final results of this study. However, due to the existence of strain specificity, there are differences in the mechanistic roles of *L. paracasei* 207-27 and BD-1 in the gut microbiota,

immune response, and mucosal barrier, which need to be further investigated subsequently.

In conclusion, our study confirms the different protective effects of short-term and long-term *L. paracasei* 207-27 administration on DSS-induced colitis with early-life antibiotics exposure. Our results suggest that short-term *L. paracasei* 207-27 administration improves the symptoms of colitis, whereas long-term *L. paracasei* 207-27 administration promotes intestinal barrier function. Our results provide a good basis for further studies on *L. paracasei* 207-27 and DSS-induced colitis.

## ACKNOWLEDGMENTS

We appreciate BYHEALTH Co., Ltd., for the bacterial powder, Enago for the English language review (assignment number: SHEXAC-9), Novogene Co., Ltd., for the 16S rRNA sequencing, and the support of Public Health and Preventive Medicine Provincial Experiment Teaching Center at Sichuan University and Food Safety Monitoring and Risk Assessment Key Laboratory of Sichuan Province.

This research was funded by the China Postdoctoral Science Foundation, grant number 2020M673267; the Full-Time Postdoctoral Research and Development Fund of Sichuan University, grant number 2020SCU12010; and the BYHEALTH Nutrition and Health Research Foundation, grant number 20H0950.

Conceptualization: X.S.; resources: N.L., F.J., Y.W., X.X., and H.L.; writing—original draft: N.L. and F.J.; writing—review and editing: Y.W. and X.S.; validation and supervision: L.L., X.Z., H.L., and X.S.; project administration: L.L., X.Z., R.C., and X.S.; funding acquisition: F.H. and X.S. All authors have read and agreed to the published version of the manuscript.

The funders had no role in the design of the study; in the collection, analysis, or interpretation of data; in the writing of the manuscript; or in the decision to publish the results.

## AUTHOR AFFILIATIONS

[1]Department of Nutrition and Food Hygiene, West China School of Public Health, West China Fourth Hospital, Sichuan University, Chengdu, China
[2]Sichuan Tianfu New Area Public Health Center, Chengdu, China
[3]BYHEALTH Institute of Nutrition & Health, Guangzhou, China

## AUTHOR ORCIDs

Niya Li http://orcid.org/0000-0003-3531-4248
Ruyue Cheng http://orcid.org/0000-0001-9333-818X
Xi Shen http://orcid.org/0009-0002-4797-0533

## FUNDING

| Funder | Grant(s) | Author(s) |
| --- | --- | --- |
| China Postdoctoral Science Foundation | 2020M673267 | Xi Shen |
| the Full-Time Postdoctoral Research and Development Found of Sichuan University | 2020SCU12010 | Xi Shen |
| the BYHEALTH Nutrition and Health Research Foundation | 20H0950 | Liang Li |

## AUTHOR CONTRIBUTIONS

Niya Li, Resources, Writing – original draft | Fengling Jiang, Resources, Writing – original draft | Yunyi Wang, Resources, Writing – review and editing | Xiaolin Xu, Resources | Liang Li, Project administration, Supervision, Validation | Xiaolei Ze, Project administration, Supervision, Validation | Huijing Liang, Resources, Supervision, Validation | Ruyue Cheng, Project administration | Fang He, Funding acquisition | Xi Shen, Funding acquisition, Project administration, Supervision, Validation, Writing – review and editing

## DATA AVAILABILITY

The 16S rRNA gene sequencing files for each feces sample have been deposited in the NCBI Sequence Read Archive (SRA) as BioProject PRJNA1254405.

## ETHICS APPROVAL

The animal study protocol was approved by the Ethics Committee of West China Fourth Hospital and West China School of Public Health of Sichuan University, under protocol Gwll2021081.

## ADDITIONAL FILES

The following material is available online.

### Supplemental Material

**Figure S1 (Spectrum02762-24-s0001.jpg).** Schematic of grouping, sample size, and intervention.
**Figure S2 (Spectrum02762-24-s0002.tiff).** Local and systemic immunities after intervention on day 21 ($n$ = 5–8).
**Figure S3 (Spectrum02762-24-s0003.tiff).** Effects on metabolites on days 21 and 46 ($n$ = 5).
**Figure S4 (Spectrum02762-24-s0004.tiff).** Local and systemic immunities after intervention on day 46 ($n$ = 5–8).
**Figure S5 (Spectrum02762-24-s0005.tiff).** Effect of a 2-week oral gavage with *Lacticasei-bacillus paracasei* 207-27 or normal saline on bacterial levels in the feces of 6-week-old BALB/c mice ($n$ = 4–5).
**Table S1 (Spectrum02762-24-s0006.xlsx).** Read number.
**Supplemental figures (Spectrum02762-24-s0007.docx).** Legends for Figures S1 to S5.

### Open Peer Review

**PEER REVIEW HISTORY (review-history.pdf).** An accounting of the reviewer comments and feedback.

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
