## [Reviewer comments · Microbiology Spectrum]

Microbiology Spectrum

Protective effect and occasion of *Lacticaseibacillus paracasei* 207-27 administration on colitis in antibiotic-exposed mice in early life

Niya Li, Fengling Jiang, Yunyi Wang, Xiaolin Xu, Liang Li, Xiaolei Ze, Huijing Liang, Ruyue Cheng, Fang He, and Xi Shen

Corresponding Author(s): Xi Shen, Sichuan University

Review Timeline:

Submission Date:	November 12, 2024
Editorial Decision:	March 7, 2025
Revision Received:	May 14, 2025
Editorial Decision:	June 5, 2025
Revision Received:	July 28, 2025
Editorial Decision:	August 13, 2025
Revision Received:	September 5, 2025
Accepted:	September 20, 2025

Editor: Bo-young Hong

Reviewer(s): Disclosure of reviewer identity is with reference to reviewer comments included in decision letter(s). The following individuals involved in review of your submission have agreed to reveal their identity: Michael G. Ganzle (Reviewer #2)

Transaction Report:

DOI: <https://doi.org/10.1128/spectrum.02762-24>

Re: Spectrum02762-24 (Protective effect and occasion of *Lactobacillus paracasei* 207-27 on colitis in antibiotic-exposed mice in early life)

Dear Dr. Xi Shen:

Thank you for the privilege of reviewing your work. Below you will find my comments, instructions from the Spectrum editorial office, and the reviewer comments.

Please carefully revise the section that describes cytokines. Specifically, lines 286-287 contain errors in the classification of pro-inflammatory and anti-inflammatory cytokines. Additionally, Fig 7N labels IL-17A on the y-axis, while the figure legend states IL-6.

Is the measurement of IFN-gamma from the colon available for day 21? It would be beneficial to expand on IFN-gamma in both the data presentation and discussion, as this cytokine plays a crucial role in inflammatory bowel disease (IBD) and colitis.

Revision Guidelines

Sincerely,
Bo-young Hong
Editor
Microbiology Spectrum

Reviewer #1 (Comments for the Author):

SUMMARY

The manuscript is well written. The introduction, methodology, results, and discussion are detailed and well presented. The authors made significant contributions to knowledge in this area of research. The conclusions are apt except for the last part which did not clearly reflect a result of investigation in this study.

The following are specific comments:

Major

1a) Importance Line 37/38. "the optimal occasion of recovery from early life antibiotic exposure is administering probiotics accompanied by antibiotic exposure".

There is no evidence in methods showing any group given antibiotics for a period of time before probiotic administration, hence no data in this study to compare with the antibiotic treatment accompanied by probiotic gavage to substantiate the above statement. Conclusions in a research article are expected to be based on results of tests carried out.

Suggestion: Consider removing the statement, rephrase it to reflect a result or results of studies performed and documented in this manuscript, or include method(s) and result(s) that justifies the statement.

1b) Discussion Line 528/529. The evidence of "our study showed that the optimal occasion is the probiotic administration accompanied by antibiotics" is not clearly demonstrated by the methods and results in this manuscript. The next sentence in line 529 reinforced this assertion, "Administering probiotics after exposure may be less effective". There is no documented evidence in the manuscript to justify the assertions since there was no data/result for antibiotic administration for a period before probiotic intervention to confirm that administering antibiotic together with probiotic (as was done in this study) gives better result than administering antibiotics without probiotics for a period before giving antibiotic with probiotic.

Consider removing it, rephrasing the statement, or including the method (s) and result(s) to justify the claim.

1c) Line 540 - 542. There is no documented proof for the conclusion "Furthermore, our study demonstrates that the optimal occasion of recovery from early life antibiotic exposure is administering probiotics accompanied by antibiotic exposure". To make this conclusion there is a need for data obtained from an investigation of a group given probiotics after a period (many hours, days, weeks, or months) of treatment with antibiotic, to be compared with data from the group treated with antibiotic accompanied by probiotics.

Suggestion: Consider removing this conclusion or rephrasing to convey what the authors have in mind based on methods and results of studies performed and documented in the manuscript. Alternatively, provide evidence with method(s) and result(s) to justify this claim.

Minor

1. Title

Consider adding the word 'administration' or any other applicable word, before the word 'on' in the title. E.g. "Protective effect and occasion of *Lactobacillus paracasei* 207-27 administration on colitis in antibiotic-exposed mice in early life".

2. Abstract

Line 17. Insert '.' after the word 'administration'

Line 18 - 20. "The first and second batches of mice were sacrificed on days 21, 46, respectively, following a 4-day intervention with 3% DSS to induce colitis". The word 'respectively' may mean that 3 % DSS was administered before day 21 and day 46 sacrifice. The 3 % DSS was administered only on day 43 before day 46 sacrifice.

Suggestion: "The first and second batches of mice were sacrificed on day 21, and day 46 following a 4-day intervention with 3 % DSS to induce colitis". Or clarify in materials and methods.

3. Introduction

Line 49/50. Remove the repeated 'compelling evidence links gut microbiota dysbiosis to IBD pathogenesis'

Line 59 and 60. Space before 'paracasei' ('L. paracasei')

4. Materials and Methods

Line 88. Change the word 'colong-forming' to 'colony-forming'

Line 103. Please change the word 'execution' to 'sacrifice'

Line 104. Define the ceftri-DSS group. Consider inserting 'the ceftri group as ceftri-DSS (n = 6). E.g., The NS group was randomly divided into an NS-water group (n = 6) and an NS-DSS group (n = 6), the ceftri group as ceftri-DSS (n = 6), and the ceftri + 207-27 group was divided into a 207-27 short-term group (n = 6) and a 207-27 long-term group (n = 6).

Line 164 - 168 are repeats of lines 158 - 162. Please delete

5. Results

Line 268. Remove '.' in word 'T.he'

Line 308/309. 'The 207-27 long-term group had a higher relative abundance of Firmicutes than the ceftri-DSS and 207-27 long-term groups'. Does this mean 'The 207-27 long-term group had a higher relative abundance of Firmicutes than the ceftri-DSS and 207-27 short-term groups'? Correct or clarify.

6. Discussion

Line 440/442. Oskul C et al. is not listed under references while Ozkul is listed in references as reference 21, not reference 26. Please change 'Oskul' to 'Ozkul', and '26' to '21'.

Line 441. Could 'non-floral flora' mean 'transient flora'? Consider using another description.

Line 446. To align with new classification, consider changing 'Lactobacillus' to 'Lacticaeibacillus'

Line 479. For easy reference identification change 'Ruyue Cheng et al.' to 'Cheng R et al.'
Line 480. For easy reference identification change 'Ying Shi et. al' to 'Shi Y et al.'
Line 482. Change 'occluding' to 'occludin'
Line 489/490. 'Short-term *L. paracasei* administration further increases'----, increases what? Please complete
Line 491. To align with new classification consider using '*Lactobacillus plantarum*' instead of '*Lactobacillus plantarum*'
Line 516. Space before 'paracasei' ('*L. paracasei*')
Supplementary Fig 1. Consider using 'sacrificed' instead of 'executed'

Reviewer #2 (Comments for the Author):

The manuscript describes experimentation to determine the probiotic intervention during early-life antibiotic treatment and during and after early-life antibiotic treatment on intestinal microbiota and the severity of DSS-induced inflammation in mice. Overall, the experimental design is relevant and experimentation is technically sound. Comments for improvement of the manuscript are indicated below.

Specific comments.

line 84/85. change to "was amplified and sequenced". Sequencing of a part of 16S rRNA gene does not provide information on strain level identity although it is useful to confirm identity of an isolate obtained from a strain collection.

line 105. For rodent experiments, the cage rather than the animal is the experimental unit and the # of cages should be indicated.

line 143. The # of sequences per sample and the bioinformatic workflow for analysis of sequences should be indicated.

line 164. A sentence should not start with a number

line 181. Reference conditions should be indicated (the NS group?)

line 198. An accession number for sequence data is missing.

Figure 2D. Is the concentration of hexanoic acid / caproic acid relevant? This is a relatively minor product of carbohydrate metabolism.

line 232. The high abundance of *Lactobacillus* is very likely a result of the probiotic intervention. It would have been preferable to quantify the probiotic strain of *L. paracasei* by strain-specific qPCR.

Figure 5B. As above, strain specific qPCR would inform whether the low abundance of *Lactobacillus* in the NS-water and Ceftri-DSS group represents the probiotic strain or other species / strains. At least data should be re-evaluated to determine whether the partial 16SrRNA gene sequences are identical to *L. paracasei*, or not.

line 412. The apparent discrepancy between long term probiotic administration and inflammatory score (Fig. 1) and its effect on microbial communities (Fig. 5) or the colonic mucosal barrier (Fig. 5) is not fully explained.

line 441 and throughout. The use of the term "flora" for microbial communities is outdated.

line 534. Experimentation described here was carried out in a mouse model for inflammatory bowel disease. A paragraph comparing intervention with *L. paracasei* to other probiotic intervention studies in the same mouse model, and to clinical trials investigating the efficacy of probiotic intervention in inflammatory bowel disease would help to bring the results in perspective.

Response to reviewers

May 6, 2025

Editorial Department of *Microbiology Spectrum*

Dear Editors and Reviewers,

Thank you for your letter and for the reviewer's comments concerning our manuscript entitled "*Lactobacillus paracasei* 207-27 on colitis in antibiotic-exposed mice in early life" (Spectrum02762-24). Those comments are all valuable and very helpful for revising and improving our paper, as well as the important guiding significance for our research. We have studied comments carefully and have made corrections which we hope meet with approval. Furthermore, we would like to show the details as follows:

Editor's comments:

- 1. Please carefully revise the section that describes cytokines. Specifically, lines 286-287 contain errors in the classification of pro-inflammatory and anti-inflammatory cytokines. Additionally, Fig. 7N labels IL-17A on the y-axis, while the figure legend states IL-6.*

The author's answer: We sincerely appreciate your careful reading. These are indeed typos, which we have corrected in the revised draft.

- 2. Is the measurement of IFN-gamma from the colon available for day 21? It would be beneficial to expand on IFN-gamma in both the data presentation and discussion, as this cytokine plays a crucial role in inflammatory bowel disease (IBD) and colitis.*

The author's answer: In our study, IFN- γ in the colon at Day 21 was measured, and the data were included in Supplement Figure 2F (see revised manuscript). As suggested, we have added a discussion on the role of IFN- γ in colitis (fifth paragraph of the discussion section). We highlight: INF- γ is one of the most severely upregulated cytokines in IBD and related mouse models^[41-43]. In this study, long-term administration of *L. paracasei* 207-27 significantly reduced colonic INF- γ expression in the DSS-induced colitis, and was more effective than short-term administration of *L. paracasei* 207-27.

Reviewer #1 (Comments for the Author):

SUMMARY

Major

1a) Importance Line 37/38. "the optimal occasion of recovery from early life antibiotic exposure is administering probiotics accompanied by antibiotic exposure".

There is no evidence in methods showing any group given antibiotics for a period of time before probiotic administration, hence no data in this study to compare with the antibiotic treatment accompanied by probiotic gavage to substantiate the above statement. Conclusions in a research article are expected to be based on results of tests carried out.

Suggestion: Consider removing the statement, rephrase it to reflect a result or results of studies performed and documented in this manuscript, or include method(s) and result(s) that justifies the statement.

The author's answer: We sincerely thank the reviewer for this critical observation. Upon re-examining our manuscript, we acknowledge that the original statement in Lines 37–38

overreaches the scope of our experimental design. We have therefore removed this sentence from the revised manuscript.

1b) Discussion Line 528/529. The evidence of "our study showed that the optimal occasion is the probiotic administration accompanied by antibiotics" is not clearly demonstrated by the methods and results in this manuscript. The next sentence in line 529 reinforced this assertion, "Administering probiotics after exposure may be less effective". There is no documented evidence in the manuscript to justify the assertions since there was no data/result for antibiotic administration for a period before probiotic intervention to confirm that administering antibiotic together with probiotic (as was done in this study) gives better result than administering antibiotics without probiotics for a period before giving antibiotic with probiotic. Consider removing it, rephrasing the statement, or including the method (s) and result(s) to justify the claim.

The author's answer: We sincerely thank the reviewer for identifying this overstatement. Upon re-evaluation, we agree that our original comparison between “probiotics accompanied by antibiotics” and “probiotics administered after antibiotic exposure” was not directly tested in the experimental design. We have therefore removed the sentences in Lines 528–529.

1c) Line 540 - 542. There is no documented proof for the conclusion "Furthermore, our study demonstrates that the optimal occasion of recovery from early life antibiotic exposure is administering probiotics accompanied by antibiotic exposure". To make this conclusion there is a need for data obtained from an investigation of a group given probiotics after a period (many

hours, days, weeks, or months) of treatment with antibiotic, to be compared with data from the group treated with antibiotic accompanied by probiotics.

Suggestion: Consider removing this conclusion or rephrasing to convey what the authors have in mind based on methods and results of studies performed and documented in the manuscript.

Alternatively, provide evidence with method(s) and result(s) to justify this claim.

The author's answer: We sincerely appreciate the reviewer's critical evaluation. Upon re-examining our manuscript, we acknowledge that the original conclusion in Lines 540–542 overextends the scope of our experimental design, as our study did not directly compare probiotic co-administration with post-antibiotic probiotic intervention. We have therefore removed this statement.

Minor

1. Title

Consider adding the word 'administration' or any other applicable word, before the word 'on' in the title. E.g. "Protective effect and occasion of Lactobacillus paracasei 207-27 administration on colitis in antibiotic-exposed mice in early life".

The author's answer: We sincerely appreciate the reviewer's thoughtful suggestion to enhance the clarity of the title. As recommended, we have added the word “administration” before “on” to explicitly indicate the intervention method. The revised title now reads: Protective effect and occasion of *Lactobacillus paracasei* 207-27 administration on colitis in antibiotic-exposed mice in early life. We confirm that the updated title has been implemented in the manuscript.

2. Abstract

Line 17. Insert '.' after the word 'administration'

The author's answer: We thank the reviewer for catching this punctuation oversight. "." has been added after "administration" in the abstract.

Line 18 - 20. "The first and second batches of mice were sacrificed on days 21, 46, respectively, following a 4-day intervention with 3% DSS to induce colitis". The word 'respectively' may mean that 3 % DSS was administered before day 21 and day 46 sacrifice. The 3 % DSS was administered only on day 43 before the day 46 sacrifice.

Suggestion: "The first and second batches of mice were sacrificed on day 21, and day 46 following a 4-day intervention with 3 % DSS to induce colitis". Or clarify in materials and methods.

The author's answer: We sincerely appreciate this critical observation. To eliminate ambiguity, we have revised the abstract as follows: The first batch of mice was sacrificed on day 21, and the second batch was sacrificed on day 46 following a 4-day intervention with 3% DSS to induce colitis. Additionally, in the Materials and Methods section (section 2.2), we have explicitly stated: "On day 21, six mice in each group were randomly selected as the first batch of mice to be sacrificed." "On day 43, except for the NS-water group, all mice in the other groups were administered with 3% DSS to induce colitis. On day 46, the experiment ended, and the remaining mice were sacrificed as the second batch."

3. Introduction

Line 49/50. Remove the repeated 'compelling evidence links gut microbiota dysbiosis to IBD pathogenesis'

The author's answer: We sincerely thank the reviewer for identifying this redundant statement. We have removed the statement in the Introduction section.

Line 59 and 60. Space before 'paracasei' ('L. paracasei')

The author's answer: We are grateful to the reviewer for noting this formatting inconsistency. We have corrected the spacing between the genus abbreviation and species name throughout the manuscript. Specifically: “*L.paracasei*” → “*L. paracasei*”. All other instances of bacterial nomenclature have been systematically checked using Word's “Find and Replace” function to ensure consistent formatting.

4. Materials and Methods

Line 88. Change the word 'colong-forming' to 'colony-forming'

The author's answer: Thank you for highlighting this error. We have corrected “colong-forming” to “colony-forming” in the revised manuscript. We have also thoroughly rechecked the text to ensure no similar spelling inconsistencies remain.

Line 103. Please change the word 'execution' to 'sacrifice'

The author's answer: Thank you for identifying this terminology inconsistency. We have revised “execution” to “sacrifice” in the updated manuscript.

Line 104. Define the ceftri-DSS group. Consider inserting 'the ceftri group as ceftri-DSS (n = 6). E.g., The NS group was randomly divided into an NS-water group (n = 6) and an NS-DSS group (n = 6), the ceftri group as ceftri-DSS (n = 6), and the ceftri + 207-27 group was divided into a 207-27 short-term group (n = 6) and a 207-27 long-term group (n = 6).

The author's answer: Thank you for your constructive feedback to improve the clarity of group definitions. We have revised the text as suggested: The NS group was randomly divided into an NS-water group and an NS–DSS group, the ceftri group as ceftri-DSS, the ceftri + 207-27 group was divided into a 207-27 short-term group and a 207-27 long-term group.

Line 164 - 168 are repeats of lines 158 - 162. Please delete

The author's answer: We sincerely appreciate the reviewer's meticulous attention to detail. The duplicated text has been removed in the revised manuscript.

5. Results

Line 268. Remove '.' in word 'The'

The author's answer: We sincerely appreciate the reviewer's meticulous attention to detail. "." has been removed in the revised manuscript.

Line 308/309. 'The 207-27 long-term group had a higher relative abundance of Firmicutes than the ceftri-DSS and 207-27 long-term groups'. Does this mean 'The 207-27 long-term group had a higher relative abundance of Firmicutes than the ceftri-DSS and 207-27 short-term groups'? Correct or clarify.

The author's answer: Thank you for highlighting this error. We have corrected in the revised manuscript (result section): The 207-27 long-term group had a higher relative abundance of *Firmicutes* than the ceftri – DSS and 207-27 short-term groups ($P < 0.01$ and 0.05 , respectively).

6. Discussion

Line 440/442. Oskul C et al. is not listed under references while Ozkul is listed in references as reference 21, not reference 26. Please change 'Oskul' to 'Ozkul', and '26' to '21'.

The author's answer: We sincerely thank the reviewers for pointing out this error. We have corrected it in the revised manuscript (discussion section).

Line 441. Could 'non-floral flora' mean 'transient flora'? Consider using another description.

The author's answer: We sincerely thank the reviewers for raising the question. We have replaced “non-floral” with “transient flora” in the revised manuscript (Discussion section).

Line 446. To align with new classification, consider changing 'Lactobacillus' to 'Lacticaseibacillus'

The author's answer: We sincerely thank the reviewers for their suggestions. We have replaced “*Lactobacillus*” with “*Lacticaseibacillus*” in the revised manuscript (Discussion section).

Line 479. For easy reference identification change 'Ruyue Cheng et al.' to 'Cheng R et al.'

The author's answer: We sincerely thank the reviewers for their suggestions. We have replaced “Ruyue Cheng et al” with “Cheng R et al” in the revised manuscript (Discussion section).

Line 480. For easy reference identification change 'Ying Shi et. al' to 'Shi Y et al.'

The author's answer: We sincerely thank the reviewers for their suggestions. We have replaced “Ying Shi et. al” with “Shi Y et al” in the revised manuscript (Discussion section).

Line 482. Change 'occluding' to 'occludin'

The author's answer: Thank you for highlighting this error. We have corrected “occluding” to “occludin” in the revised manuscript. We have also thoroughly rechecked the text to ensure no similar spelling inconsistencies remain.

Line 489/490. 'Short-term L. paracasei administration further increases'----, increases what?

Please complete

The author's answer: We sincerely thank the reviewers for raising the question. We have completed the revised manuscript: The greater elevation of sIgA after short-term *L. paracasei* 207-27 administration, possibly because of a facilitating effect even after the inhibition of pathogenic bacteria.

Line 491. To align with the new classification consider using 'Latilactobacillus plantarum' instead of 'Lactobacillus plantarum'

The author's answer: We sincerely thank the reviewers for the important updates to ensure taxonomic accuracy. We have revised “*Lactobacillus plantarum*” to “*Latilactobacillus plantarum*” in the revised manuscript.

Line 516. Space before 'paracasei' ('L. paracasei')

The author's answer: We sincerely appreciate the reviewer's meticulous attention to detail. We have added a space before "paracasei" .

Supplementary Fig 1. Consider using 'sacrificed' instead of 'executed'

The author's answer: Thank you for identifying this terminology inconsistency. We have revised "executed" to "sacrificed" in the updated manuscript.

Reviewer #2 (Comments for the Author):

Specific comments.

1. *line 84/85. change to "was amplified and sequenced". Sequencing of a part of 16S rRNA gene does not provide information on strain level identity although it is useful to confirm identity of an isolate obtained from a strain collection.*

The author's answer: We sincerely appreciate the reviewer's insightful feedback to enhance methodological precision. The sentence has been revised as suggested: To confirm bacterial strain at the species level, V3–V4 of 16S rRNA sequencing was amplified and sequenced.

2. *line 105. For rodent experiments, the cage rather than the animal is the experimental unit and the # of cages should be indicated.*

The author's answer: We sincerely thank the reviewer for emphasizing this critical statistical consideration. We have revised the manuscript to explicitly clarify the experimental unit and

provide cage-level details as follows: The number of cages per group was 1, with 6 mice per cage.

3. *line 143. The # of sequences per sample and the bioinformatic workflow for analysis of sequences should be indicated.*

The author's answer: We sincerely appreciate the reviewer's suggestion to enhance the methodological clarity. We have now added the following details to the revised manuscript (Methods Section 2.4): The average number of raw sequences per sample was 62625-97171. The detailed sequence counts per sample are provided in Supplementary Table 1.

4. *line 164. A sentence should not start with a number*

The author's answer: We thank the reviewer for highlighting this stylistic oversight. This paragraph has been deleted in the revised version due to repetition.

5. *line 181. Reference conditions should be indicated (the NS group?)*

The author's answer: We thank the reviewers for emphasizing the need for explicit reporting of the experimental design. We have made the following additions to clarify the reference conditions: the ceftri group as the control group on day 21, and the NS-water group as the control group on day 46.

6. *line 198. An accession number for sequence data is missing.*

The author's answer: We sincerely apologize for this oversight and appreciate the opportunity

to clarify data accessibility. The sequence data associated with this study have been deposited in NCBI under the following accession numbers: PRJNA1254405.

7. *Figure 2D. Is the concentration of hexanoic acid / caproic acid relevant? This is a relatively minor product of carbohydrate metabolism.*

The author's answer: Thank you for raising this critical point. We agree that hexanoic acid /caproic acid is typically a minor metabolite in canonical carbohydrate metabolism pathways. However, in our previous studies, we found that hexanoic acid is associated with intestinal immune function and can regulate Th17/Treg balance, and hexanoic acid is associated with intestinal barrier function and can reduce inflammation levels. So, although these two are not major short-chain fatty acids, we still tested them.

8. *line 232. The high abundance of Lacticaseibacillus is very likely a result of the probiotic intervention. It would have been preferable to quantify the probiotic strain of L. paracasei by strain-specific qPCR.*

The author's answer: We thank the reviewer for raising this important methodological consideration. We acknowledge that strain-specific qPCR could provide precise quantification of the administered *L. paracasei* strain. Unfortunately, strain-specific PCR validation is not possible at this time because, technically, strain-specific primers for *L. paracasei* 207-27 had not been developed. The experimental results will be predicted later when primers are available. We also speculate that the high abundance of *Lacticaseibacillus* is related to the intervention of *L. paracasei* 207-27, and we also expect the bacterium to stay and colonize.

9. *Figure 5B. As above, strain specific qPCR would inform whether the low abundance of Lacticaseibacillus in the NS-water and Ceftri-DSS group represents the probiotic strain or other species / strains. At least data should be re-evaluated to determine whether the partial 16SrRNA gene sequences are identical to L. paracasei, or not.*

The author's answer: We sincerely appreciate this critical observation. The distinction between the administered probiotic strain (*L. paracasei* 207-27) and endogenous *Lacticaseibacillus* species is indeed crucial for interpreting colonization dynamics. Unfortunately, strain-specific PCR validation is not possible at this time because, technically, strain-specific primers for *L. paracasei* 207-27 had not been developed. The experimental results will be predicted later when primers are available. As to whether the 16S rRNA gene sequence is identical to that of *L. paracasei*, we make the following explanation: At the time of this experiment, an old database was used with the taxonomic name *Lactobacillus paracasei*, and although the name was subsequently adjusted taxonomically to *Lacticaseibacillus*, the sequencing results for *Lacticaseibacillus* in this experiment corresponded to *L. paracasei*.

10. *line 412. The apparent discrepancy between long term probiotic administration and inflammatory score (Fig. 1) and its effect on microbial communities (Fig. 5) or the colonic mucosal barrier (Fig. 5) is not fully explained.*

The author's answer: We appreciate the reviewer's insightful observation. The relationship between probiotic duration, inflammation modulation, and the colonic mucosal barrier is indeed complex. We have expanded the Discussion section (paragraph 2): This may be because

inflammation relief and intestinal flora adjustment require only short-term probiotic colonization, whereas restoration of the intestinal barrier requires long-term sustained probiotic colonization.

11. line 441 and throughout. The use of the term "flora" for microbial communities is outdated.

The author's answer: We sincerely thank the reviewer for highlighting this important terminology oversight. We fully acknowledge that the term "flora" is no longer aligned with current scientific nomenclature for microbial communities. We have replaced it in the revised manuscript: Ozkul C et al. suggested that long-term antibiotic administration can decrease the dominant microbial communities and an overgrowth of foreign or transient microbial communities, which may lead to a further increase in alpha diversity^[21].

12. lien 534. Experimentation described here was carried out in a mouse model for inflammatory bowel disease. A paragraph comparing intervention with L. paracasei to other probiotic intervention studies in the same mouse model, and to clinical trials investigating the efficacy of probiotic intervention in inflammatory bowel disease would help to bring the results in perspective.

The author's answer: We thank the reviewer for this constructive suggestion. We have now added a dedicated paragraph in the Discussion section (paragraph 6) to contextualize our findings within existing evidence: PENG C et al. found that early intervention with BD-1 and BD-1 + ceftri had a protective and preventive effect in a DSS-induced colitis model, as evidenced by a reduction in inflammation scores and MPO activity levels^[18]. This is consistent with the final results of this study. However, due to the existence of strain specificity, there are

differences in the mechanistic roles of *L. paracasei* 207-27 and BD-1 in the gut microbiota, immune response, and mucosal barrier, which need to be further investigated subsequently.

Thank you very much for your attention and time. Look forward to hearing from you.

Yours Sincerely.

Corresponding author: Xi Shen

Institution and address: Department of Nutrition and Food Hygiene, West China

School of Public Health and West China Fourth Hospital, Sichuan University,

Chengdu, 610041, China

E-mail: hxgwshenxi@sina.com

Tel.: +86-152-0821-3847

Re: Spectrum02762-24R1 (Protective effect and occasion of *Lactobacillus paracasei* 207-27 administration on colitis in antibiotic-exposed mice in early life)

Dear Dr. Xi Shen:

Thank you for the privilege of reviewing your work. Below you will find my comments, instructions from the Spectrum editorial office, and the reviewer comments.

Please respond to the reviewer's thorough and constructive feedback to improve the manuscript. Specifically Fig5b to provide clearer data representation and ensure it aligns with the findings. Additionally reevaluate study design regarding the number of cage included in the analysis to ensure rigorous methodology and robust results.

Revision Guidelines

Sincerely,
Bo-young Hong
Editor
Microbiology Spectrum

Reviewer #2 (Comments for the Author):

Title and throughout. Current nomenclature should be used (*Lactobacillus paracasei*)
Section 2.2. Mice are coprophagic; for microbiome-targeted experimentation, the cage rather than the animal is the experimental

unit. With $n=1$, the replication is somewhat underwhelming and it is impossible to differentiate between cage effects and treatment effects.

Section 2.4. Short sequences of 16S rRNA gene amplicons do not inform on species-level taxonomy.

Figure 5B makes no sense. *Lactocaseibacillus* is not an abundant member of microbiota of mice, DSS-treated or not, so we would expect sequences matching the genus only if the organism was gavaged as probiotic organism. Here, the abundance is highest in two groups that did not receive *Lc. paracasei* as a probiotic but lowest in the one of the two groups that did. As the experiment was not replicated there may be an element of random here.

Strain-specific quantification of the probiotic strain would have been of great help. Several protocols to design strain specific primers for probiotic strains to achieve absolute quantification in intestinal samples were published in the past 5 years (e.g. <https://doi.org/10.1186/s40168-024-01881-2>).

Response to reviewers

July 20, 2025

Editorial Department of *Microbiology Spectrum*

Dear Editors and Reviewers,

Thank you for your letter and for the reviewer's comments concerning our manuscript entitled "Protective effect and occasion of *Lacticaseibacillus paracasei* 207-27 administration on colitis in antibiotic-exposed mice in early life" (Spectrum02762-24R1). Those comments are all valuable and very helpful for revising and improving our paper, as well as the important guiding significance for our research. We have studied comments carefully and have made corrections which we hope meet with approval. Furthermore, we would like to show the details as follows:

Reviewer #2 (Comments for the Author):

1. *Title and throughout. Current nomenclature should be used (Lacticaseibacillus paracasei)*

The author's answer: We sincerely thank the reviewer for highlighting this important point. We have now updated all instances of the previous designation (*Lactobacillus paracasei*) to the currently accepted nomenclature (*Lacticaseibacillus paracasei*) throughout the manuscript, including the title, abstract, main text, figures, tables, and supplementary materials.

2. *Section 2.2. Mice are coprophagic; for microbiome-targeted experimentation, the cage rather than the animal is the experimental unit. With n=1, the replication is somewhat underwhelming and it is impossible to differentiate between cage effects and treatment effects.*

The author's answer: We sincerely thank the reviewer for this essential critique. We fully agree

that mice treating the cage as the experimental unit in microbiome studies. There are problems with the formulation of grouping and sample size in the first draft of the paper, and we apologize for this oversight. We went and re-verified and reconfirmed the original records and are available if you need them. The details are as follows: Fourteen-day-old timed-pregnant BALB/C female mice were purchased at the beginning of the experiment, and after the pregnant mice gave birth to mice, 42 neonatal mice were randomly selected on day 0 and divided into NS, NS-ceftri, and ceftri+207-27 groups, and there were 18 mice in the NS group (6 mice/cage, 3 cages), 12 mice in the NS-ceftri group (6 mice/cage, 2 cages), 18 mice in the ceftri+207-27 group (6 mice/cage, 3 cages), with every 6 mice in a cage with the mother. After weaning at the end of the intervention at week 3, 2 mice per cage in the NS group were randomly sacrificed, and the remaining mice continued to be divided into NS-water and NS-DSS groups, at which time there were 6 mice in each group (2 mice/cage, 3 cages); 3 mice per cage in the NS-ceftri group were randomly sacrificed, and the remaining mice were in the ceftri-DSS group (3 mice/cage, 2 cages); and 3 mice per cage in the ceftri+207-27 group was randomly sacrificed with 2 mice per cage, and the remaining mice continued to be divided into 207-27 short-term and 207-27 long-term groups (2 mice/cage, 3 cages). Due to the complexity of the experimental design, we denote the number of mice in each group by N. We have made corrections and additions in section 2.2 of the manuscript.

3. *Section 2.4. Short sequences of 16S rRNA gene amplicons do not inform on species-level taxonomy.*

The author's answer: We thank the reviewer for this critical methodological observation, which

we fully acknowledge as a errors in terminology, 16S rRNA sequencing differentiates and identifies at the phylum and genus level and does not identify at the species level, we have been corrected in the manuscript.

4. *Figure 5B makes no sense. Lacticaseibacillus is not an abundant member of microbiota of mice, DSS-treated or not, so we would expect sequences matching the genus only if the organism was gavaged as probiotic organism. Here, the abundance is highest in two groups that did not receive Lc. paracasei as a probiotic but lowest in the one of the two groups that did. As the experiment was not replicated to there may be an element of random here.*

The author's answer: We thank the reviewer for keen observation. We explained this in detail: Figure 5B shows the microbiota data at the end of the experiment, in which the NS-water and NS-DSS groups were not given antibiotics and the strains were not given by gavage, so the measured *lacticaseibacillus* levels are the natural level in the mice, which are really not very high in terms of relative abundance and number (0.01-0.02%). The 207-27 short-term group was gavaged with both antibiotics and bacteria for 3 weeks. Because the antibiotics killed the original *lacticaseibacillus*, the level of *lacticaseibacillus* was lower in this group after discontinuing the gavage, and was slightly higher in this group compared to the control group (ceri-DSS), which was also used antibiotics, but there was no difference in *P* value. However, the 207-27 long-term group continued to gavage *L. paracasei* 207-27 up to 6 weeks after discontinuing antibiotics on 3 weeks, so the abundance of *Lactobacillus* remained high at the end of the experiment. In addition, as can be seen in Figure 2B, when the antibiotic and *L. paracasei* 207-27 were administered by gavage together for 3 weeks, the abundance of *lacticaseibacillus* actually increased compared to

the antibiotic group, which suggests that the *L. paracasei* 207-27 that we gavaged entered the intestine after the antibiotics had killed the intestinal native bacteria. We have added details and marked yellow in the discussion section.

5. *Strain-specific quantification of the probiotic strain would have been of great help. Several protocols to design strain specific primers for probiotic strains to achieve absolute quantification in intestinal samples were published in the past 5 years (e.g. <https://doi.org/10.1186/s40168-024-01881-2>).*

The author's answer: We sincerely thank the reviewer for this valuable suggestion. We fully agree that strain-specific quantification of the administered probiotic strain in intestinal samples would have provided deeper insights into colonization dynamics and host-microbe interactions. But the detection of the strain level was indeed limited and the strain primers were not disclosed. Therefore, we performed strain quantification based on the literature that addresses quantitative testing of this strain (<https://doi.org/10.3389/fmicb.2024.1456274>). The results showed that *L. paracasei* 207-27 entered the intestinal tract of mice after direct gavage compared to the blank control group. This data was added to the additional material.

The concentration of *Lacticaseibacillus paracasei* (CFU/g)

control	207-27
3511.658	169077.4
13376.14	142216.6
14000.79	345687.6
244.49	653381.4
496675.9	

Re: Spectrum02762-24R2 (Protective effect and occasion of *Lactocaseibacillus paracasei* 207-27 administration on colitis in antibiotic-exposed mice in early life)

Dear Dr. Xi Shen:

Thank you for the privilege of reviewing your work. Below you will find my comments, instructions from the Spectrum editorial office, and the reviewer comments.

Please respond to the reviewer's comments with careful attention, ensuring that each point is adequately addressed. Furthermore, it is essential to provide a detailed account of the modifications made to the study design in response to the reviewer's suggestions since the initial submission. This response should demonstrate a clear understanding of the feedback and illustrate how it has been integrated to enhance the quality and rigor of the study.

Revision Guidelines

Sincerely,
Bo-young Hong
Editor
Microbiology Spectrum

Reviewer #2 (Comments for the Author):

The manuscript has improved with the manuscript revision but I remain with three major comments:

- section 2.2. "mice and treatments". A replication with 2 or 3 cages per group remains underwhelming but comparable publications use similar numbers and my expertise in animal experiments is not sufficient to judge how to properly design such a rather complex animal experiments. To add insult to injury, the experimental design not only should account for cage effects but also for litter effects (which do matter in experiments with newborn animals). The section is rather long and convoluted; it would help to depict the experimental design in a figure. As sufficient replication is at least questionable, conclusions related to the intestinal microbiota should be qualified and condensed.

- the primers used to generate Figure S5 are species specific, not strain specific. This is still useful and increases the confidence that the probiotic strain was quantified but this should be indicated. The methods used to generate the figure must be indicated in the main manuscript and the legend must be improved to understand what was done and which samples were used. It appears that none of the supplementary figures are referred to in the main manuscript - either the reference to the supplementary figures are included in the main manuscript (along with a full description of the methods in the main manuscript) or the supplementary figures should be deleted. I have a strong preference for keeping the supplementary figures, particularly Figure S5.

Minor comments:

line 55. I trust that the isolates were obtained from fecal samples, not from the intestine?

line 95. The sentence starting with "and there were" is hanging.

lines 344 and 346. "Error reference not found" should be corrected.

line 435 and throughout. Genus names are capitalized.

line 437 "which were also treated with antibiotics" is meant? Mice generally don't "use" antibiotics.

Response to reviewers

August 28, 2025

Editorial Department of *Microbiology Spectrum*

Dear Editors and Reviewers,

Thank you for your letter and for the reviewer's comments concerning our manuscript entitled "Protective effect and occasion of *Lacticaseibacillus paracasei* 207-27 administration on colitis in antibiotic-exposed mice in early life" (Spectrum02762-24R2). Those comments are all valuable and very helpful for revising and improving our paper, as well as the important guiding significance for our research. We have studied comments carefully and have made corrections which we hope meet with approval. Furthermore, we would like to show the details as follows:

Reviewer #2 (Comments for the Author):

1. *section 2.2. "mice and treatments". A replication with 2 or 3 cages per group remains underwhelming but comparable publications use similar numbers and my expertise in animal experiments is not sufficient to judge how to properly design such a rather complex animal experiments. To add insult to injury, the experimental design not only should account for cage effects but also for litter effects (which do matter in experiments with newborn animals). The section is rather long and convoluted; it would help to depict the experimental design in a figure. As sufficient replication is at least questionable, conclusions related to the intestinal microbiota should be qualified and condensed.*

The author's answer: We sincerely thank the reviewer for this exceptionally insightful and critical comment, which highlights the fundamental challenges in designing robust animal experiments, particularly those involving microbiota. We agree entirely with the reviewer's

assessment and have revised the manuscript accordingly to address these concerns in a comprehensive manner. Firstly, regarding the number of animals in each group and the independence of the samples, it is true that similar studies have adopted similar operational methods and groupings, and we herein describe the experimental procedure in greater detail. As for the litter effect, we have randomized to the greatest extent possible during our experiments. As mentioned in the Section 2.2, newborn mice were not always kept in the same cage with their mothers and entered into the same intervention group, but were randomly assigned to different cages, and each cage was randomly assigned with feeding mothers to mitigate the litter effect. Admittedly, the cage effect did exist, and because of space constraints, we ended up with only 2-3 cage replicates per intervention group, but we also considered the effect of different cages when collecting samples in batches, randomized the combinations, and employed a random selection process where mice were sacrificed across two separate batches to further minimize the potential confounding cage effect. Given that our experiments may indeed have been influenced by cage effects on colonization, as you say, we should have been more cautious in our colonization-related conclusions, and this has been added at the end of the third paragraph of the discussion. Finally, we also added an illustration of the experimental flow in Supplement Figure 1 to present the experimental process more clearly.

Supplementary Fig 1 Schematic of Grouping, Sample Size, and Intervention

2. *the primers used to generate Figure S5 are species specific, not strain specific. This is still useful and increases the confidence that the probiotic strain was quantified but this should be indicated. The methods used to generate the figure must be indicated in the main manuscript and the legend must be improved to understand what was done and which samples were used.*

The author's answer: We are grateful to the reviewer for raising this important point. We fully agree with the comment and have thoroughly revised the manuscript to address it as follows:

a) Section 2.5: Validation of bacterial strain entry into the intestine via oral gavage

The 6-week-old BALB/c mice were divided into two groups, 5 in each group, and were

given the same dose of *L. paracasei* 207-27 bacterial solution and normal saline respectively for 2 weeks. Collected fresh feces and extracted DNA, using *L. paracasei* 207-27 species primers and methods verified by Guo, L et al^[20], and performed real-time PCR to determine the abundance level of *Lacticaseibacillus paracasei* in mouse feces, and obtained Supplementary Figure 5.

- b) **Section 3.2:** It has been confirmed that the bacterial strain reached the intestine via oral gavage and can be detected in fecal samples, indicating its ability to tolerate the digestive process and potentially colonize the gut to modulate the gut microbiota, as shown in Supplementary Figure 5.
- c) **Figure Legend:** The legend for **Supplementary Figure S5** has been completely rewritten and significantly improved: Effect of a 2-week oral gavage with *Lacticaseibacillus paracasei* 207-27 or normal saline on bacterial levels in the feces of 6-week-old BALB/c mice (n=4-5).

We believe these revisions have greatly enhanced the transparency of our methods and the clarity of our results. We thank the reviewer again for this helpful suggestion.

- 3. *It appears that none of the supplementary figures are referred to in the main manuscript - either the reference to the supplementary figures are included in the main manuscript (along with a full description of the methods in the main manuscript) or the supplementary figures should be deleted. I have a strong preference for keeping the supplementary figures, particularly Figure S5.*

The author's answer: We thank the reviewer for this crucial observation. We entirely agree that

supplementary materials must be referenced in the main text to be valid, and we apologize for this oversight. Following the reviewer's strong recommendation to keep the figures (especially Supplementary Fig. S5), we have now inserted citations for all supplementary figures in the appropriate sections of the main manuscript.

4. *line 55. I trust that the isolates were obtained from fecal samples, not from the intestine?*

The author's answer: We thank the reviewer for catching this important terminology error. We have replaced "*L. paracasei* 207-27 was isolated from the intestine of healthy infants" with "*L. paracasei* 207-27 was isolated from healthy infant feces". The change has been made on Introduction and are also highlighted in yellow in revised manuscript.

5. *line 95. The sentence starting with "and there were" is hanging.*

The author's answer: We thank the reviewer for pointing out this grammatical error. We have rewritten the sentence to improve its structure and clarity. The revised sentence now reads: "The mice were housed with their mothers at a density of six pups per cage, resulting in three cages for the NS group, two cages for the ceftri group, and three cages for the ceftri + 207-27 group." .

6. *lines 344 and 346. "Error reference not found" should be corrected.*

The author's answer: We thank the reviewer for noticing this error. The missing references have been identified and properly re-inserted into the manuscript.

7. *line 435 and throughout. Genus names are capitalized.*

The author's answer: We thank the reviewer for this critical reminder regarding microbiological nomenclature. We apologize for this oversight. We have now conducted a full, word-by-word review of the entire manuscript to ensure that all genus names are correctly capitalized. This check included the Abstract, Main Text, Figures, and Figure Legends. All identified errors have been corrected.

8. *line 437 "which were also treated with antibiotics" is meant? Mice generally don't "use" antibiotics.*

The author's answer: We thank the reviewer for catching this error. We have replaced “used” with “treated”. The change has been made on Discussion and are also highlighted in yellow in revised manuscript.

Re: Spectrum02762-24R3 (Protective effect and occasion of *Lactocaseibacillus paracasei* 207-27 administration on colitis in antibiotic-exposed mice in early life)

Dear Dr. Xi Shen:

Please follow the instruction below.

Your manuscript has been accepted, and I am forwarding it to the ASM production staff for publication. Your paper will first be checked to make sure all elements meet the technical requirements. ASM staff will contact you if anything needs to be revised before copyediting and production can begin. Otherwise, you will be notified when your proofs are ready to be viewed.

Sincerely,
Bo-young Hong
Editor
Microbiology Spectrum

Reviewer #2 (Comments for the Author):

Preferably, replace "real-time PCR" by "quantitative PCR" but this can be done at the proof stage.